# A toolbox of nanobodies developed and validated for use as intrabodies and nanoscale immunolabels in mammalian brain neurons

Jie-Xian Dong[1], Yongam Lee[1], Michael Kirmiz[1], Stephanie Palacio[1], Camelia Dumitras[1], Claudia M Moreno[2,3], Richard Sando[4], L Fernando Santana[2], Thomas C Südhof[4], Belvin Gong[1], Karl D Murray[1], James S Trimmer[1,2]*

[1]Department of Neurobiology, Physiology and Behavior, University of California, Davis, Davis, United States; [2]Department of Physiology and Membrane Biology, University of California, Davis, Davis, United States; [3]Department of Physiology and Biophysics, University of Washington, Seattle, United States; [4]Department of Molecular and Cellular Physiology, Howard Hughes Medical Institute, Stanford School of Medicine, Stanford, United States

**Abstract** Nanobodies (nAbs) are small, minimal antibodies that have distinct attributes that make them uniquely suited for certain biomedical research, diagnostic and therapeutic applications. Prominent uses include as intracellular antibodies or intrabodies to bind and deliver cargo to specific proteins and/or subcellular sites within cells, and as nanoscale immunolabels for enhanced tissue penetration and improved spatial imaging resolution. Here, we report the generation and validation of nAbs against a set of proteins prominently expressed at specific subcellular sites in mammalian brain neurons. We describe a novel hierarchical validation pipeline to systematically evaluate nAbs isolated by phage display for effective and specific use as intrabodies and immunolabels in mammalian cells including brain neurons. These nAbs form part of a robust toolbox for targeting proteins with distinct and highly spatially-restricted subcellular localization in mammalian brain neurons, allowing for visualization and/or modulation of structure and function at those sites.

DOI: https://doi.org/10.7554/eLife.48750.001

*For correspondence:
jtrimmer@ucdavis.edu

Competing interests: The authors declare that no competing interests exist.

## Introduction

Nanobodies (nAbs) are the recombinant minimal antigen binding fragments derived from the atypical monomeric immunoglobulins present in camelid mammals and cartilaginous fish (*Hamers-Casterman et al., 1993*; *Muyldermans, 2013*; *Desmyter et al., 2015*; *Beghein and Gettemans, 2017*; *Könning et al., 2017*; *De Meyer et al., 2014*). They have broad utility as biomedical research reagents, diagnostics and therapeutics. nAbs are ideally suited for use as intracellular antibodies (intrabodies) in living cells (*Lafaye et al., 2009*; *Beghein et al., 2016*; *Staus et al., 2014*; *Bertier et al., 2017*; *Van Audenhove and Gettemans, 2016*; *Schumacher et al., 2018*), as they fold efficiently and remain stable under a wide range of conditions, including the reducing cytoplasmic environment (*Gahrtz and Conrad, 2009*; *Böldicke et al., 2005*; *Goenaga et al., 2007*; *Lynch et al., 2008*). In addition to their potential utility as intrabodies, nAbs also have advantages as immunolabeling reagents, as their small size ($\approx 1/10$ of conventional IgG antibodies) improves penetration of the cell or tissue samples (*Perruchini et al., 2009*; *Fang et al., 2018*). Importantly, nAbs improve imaging resolution by reducing the distance between the immunolabeling signal and the

target protein from the 10–20 nm obtained with conventional primary and secondary antibodies down to 2–4 nm (*Beghein and Gettemans, 2017*; *Ries et al., 2012*; *Szymborska et al., 2013*; *Pleiner et al., 2015*). Their ability to precisely target specific proteins in living cells, and/or label them in *post vivo* samples with a high degree of efficacy and spatial resolution make nAbs attractive for numerous biomedical research applications, including in neuroscience research (*Südhof, 2018*). However, their uses in neuroscience have largely been limited to samples exogenously expressing GFP-tagged proteins [*e.g.*, (*Fang et al., 2018*; *Ekstrand et al., 2014*; *Tang et al., 2013*; *Chamma et al., 2016*; *Joensuu et al., 2016*; *Modi et al., 2018*)], or in studies targeting proteins expressed in non-neuronal brain cells (*Fang et al., 2018*), although a set of recent studies have employed nAbs against neuronal targets [*e.g.*, (*Schoonaert et al., 2017*; *Schenck et al., 2017*; *Scholler et al., 2017*; *Maidorn et al., 2019*).

Mammalian brain neurons are distinguished from other cells by extreme molecular and structural complexity that is intimately linked to the array of intra- and inter-cellular signaling events that underlie brain function. Integral to the functional complexity of neurons is the diversity of proteins they express (estimated to encompass the products of two-thirds of the genome), a complexity markedly enhanced by compartmentalization of specific proteins at highly restricted sites within the neuron's complex structure. This includes not only the basic polarized compartments (dendrite, cell body, axon), but also distinct subcompartments within these domains (*e.g.*, dendritic spines, the axon initial segment [AIS], nodes of Ranvier, presynaptic terminals, etc.). Each of these sites is responsible for distinct events in neuronal signaling and function, creating opportunities for specific delivery of reporters and actuators to these sites with high subcellular resolution to report on or influence, respectively, specific aspects of neuronal function.

Here, we describe the development and characterization of recombinant nAbs with specificity for a set of neuronal proteins with restricted expression in subcellular compartments associated with discrete signaling events crucial to mammalian brain neuron function and plasticity. These targets are the postsynaptic scaffolding proteins Homer1 (*Brandstätter et al., 2004*), IRSp53 (*Soltau et al., 2002*), and SAPAP2 (*Takeuchi et al., 1997*) that are present at partially overlapping sets of excitatory synapses, Gephyrin (*Kneussel et al., 2001*) found postsynaptically at most inhibitory synapses, and the Kv2 channel auxiliary subunit AMIGO-1 (36) found in large clusters at endoplasmic reticulum-plasma membrane (ER-PM) junctions present on the soma, proximal dendrites and AIS.

## Results

### Generation of nAbs against brain target proteins and their isolation by panning phage display libraries prepared from an immunized llama

Neuronal proteins with a restricted localization in specific subcellular compartments were targeted for nAb development (*Figure 1*). We isolated lymphocytes from a single llama immunized with recombinant fragments of these five target proteins and generated nAb phage display cDNA libraries that were subsequently used to isolate target-specific nAbs *via* phage binding to the individual target proteins. After verifying target specificity by ELISA, we sequenced and evaluated unique ELISA-positive nAbs for use as intrabodies in heterologously expressing mammalian cells and in cultured hippocampal neurons expressing endogenous target proteins. We also evaluated unique ELISA-positive nAbs for use as immunolabels for immunofluorescence immunocytochemistry (IF-ICC) on heterologous cells, immunohistochemistry (IHC) on rat brain sections, and immunoblots (IB) on crude rat brain membranes. This stepwise screening approach (*Figure 1*) led to the identification of novel nAbs for use as intrabodies and as immunolabels (*Table 1*).

The immunogens used were recombinant fragments of intracellular domains from a strategically chosen set of five neuronal protein targets involved in synaptic signaling and neuronal excitability (Homer1, IRSp53, SAPAP2, Gephyrin and AMIGO-1). Recombinant protein fragments were produced and purified from *E. coli* and combined in a cocktail to immunize a single llama (*Figure 1*). Antiserum was collected at intervals and assayed for immunoreactivity against the separate recombinant protein fragments by ELISA. We also purified IgG fractions corresponding to the conventional heavy and light chain subclass versus the heavy chain-only subclasses and assayed them for immunoreactivity against the individual target proteins by ELISA. Once a sufficient titer was achieved, we obtained whole blood and isolated the leukocytes to use as source of total RNA that served as a

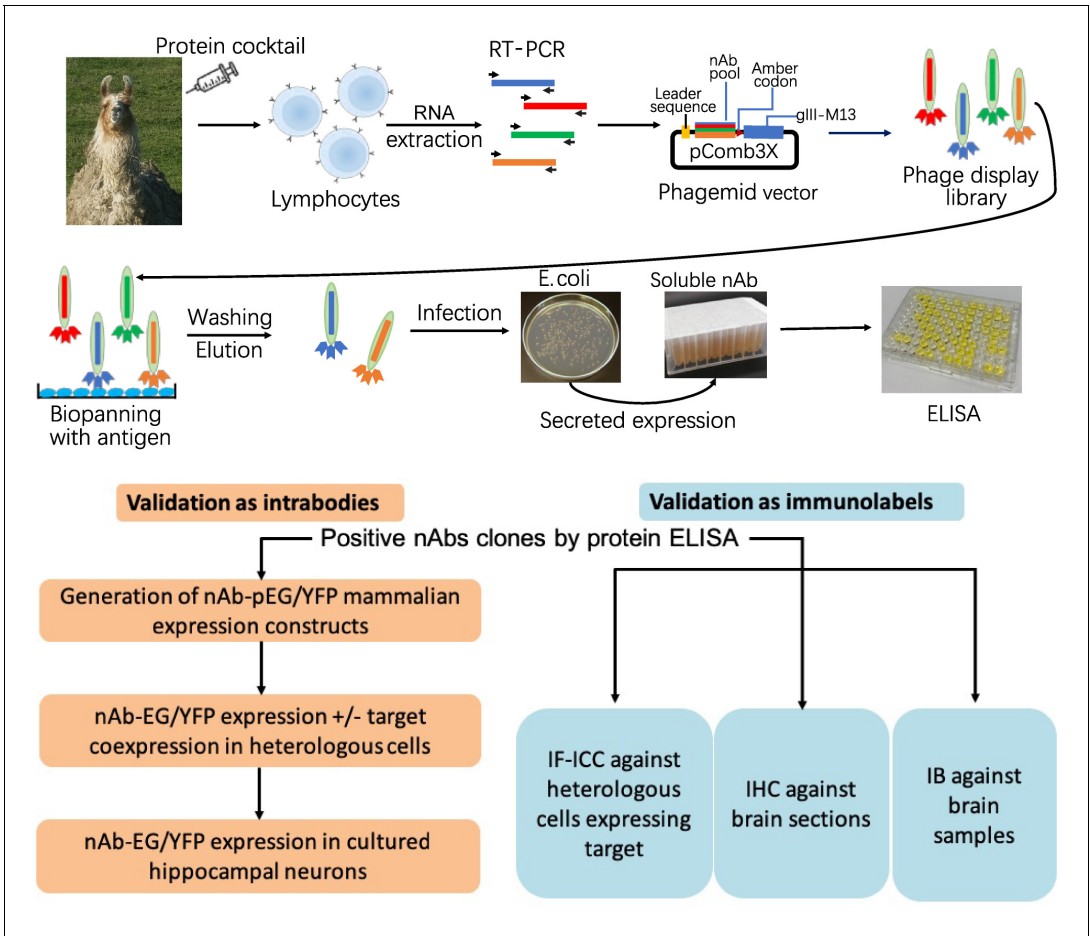

**Figure 1.** Schematic of nAb generation and validation pipeline. (**A**) Schematic of nAb generation pipeline. See text for details. (**B**) Schematic of pipeline for validating ELISA-positive nAbs for utility as intrabodies and immunolabels.
DOI: https://doi.org/10.7554/eLife.48750.002

The following figure supplements are available for figure 1:

**Figure supplement 1.** Anti-Homer1 nAbs that function as intrabodies colocalize with exogenously expressed Homer1 when coexpressed in heterologous COS-1 cells.
DOI: https://doi.org/10.7554/eLife.48750.003

**Figure supplement 2.** nAbs that function as intrabodies against excitatory synaptic target proteins colocalize with the endogenously expressed target proteins in cultured hippocampal neurons.
DOI: https://doi.org/10.7554/eLife.48750.004

**Figure supplement 3.** nAbs that function as intrabodies against inhibitory synaptic and ER-PM junction target proteins colocalize with endogenously expressed target proteins in cultured hippocampal neurons.
DOI: https://doi.org/10.7554/eLife.48750.005

**Figure supplement 4.** nAbs that function as intrabodies against excitatory synaptic target proteins colocalize with the excitatory synaptic marker PSD-95 in cultured hippocampal neurons.
DOI: https://doi.org/10.7554/eLife.48750.006

**Figure supplement 5.** nAbs that function as intrabodies against inhibitory synaptic and ER-PM junction target proteins localize to their respective subcellular domains in cultured hippocampal neurons.
DOI: https://doi.org/10.7554/eLife.48750.007

**Figure supplement 6.** Anti-Homer1 nAbs exhibit labeling of exogenously expressed Homer1 in heterologous COS-1 cells.
DOI: https://doi.org/10.7554/eLife.48750.008

template for RT-PCR to specifically amplify cDNAs corresponding to the IgG heavy-chain variable fragments. These cDNAs were ligated into the pComb3XSS phagemid which allows for expression of the cDNA-encoded nAbs as PIII protein fusions on the pili of phage (*Andris-Widhopf et al., 2000*).

**Table 1.** Summary of nanobody generation and validation.

| Target | Primary selection and validation | | | Validation as intrabodies | | Validation as immunolabels | | |
|---|---|---|---|---|---|---|---|---|
| | Phage clones selected | ELISA positives | Unique ELISA positives | COS-1 intrabody positives | Neuron intrabody positives | COS-1 IF-ICC positives | Brain IHC positives | Brain IB positives |
| Homer1 | 180 | 135 | 39 | 32 | 12 | 33 | 25 | 13 |
| IRSp53 | 160 | 33 | 17 | 8 | 3 | 0 | 0 | 0 |
| SAPAP2 | 172 | 32 | 15 | 7 | 2 | 0 | 4 | 0 |
| Gephyrin | 182 | 78 | 24 | 9 | 5 | 0 | 1 | 2 |
| AMIGO-1 | 173 | 38 | 18 | 13 | 5 | 0 | 0 | 0 |

DOI: https://doi.org/10.7554/eLife.48750.009

We subjected this phage display library (complexity $\approx 8 \times 10^7$) to panning against the individual purified recombinant target proteins immobilized in wells of a microtiter plate. Bound and eluted phage were assayed by phage ELISA against the individual target proteins, which demonstrated substantial enrichment of immunoreactivity after one round of panning (not shown). We also used 'sandwich' panning, in which wells of a microtiter plate were coated with pooled IgG fractions purified from the immunized llama to capture and display the individual recombinant target protein fragments. The bound and eluted phage from sandwich panning were subjected to two more rounds of sandwich panning. *Table 1* contains a summary of the outcome of the phage display.

Isolated phage were used to infect fresh cultures of *E. coli* bacteria ER2738 which were plated onto bacterial culture plates, from which single colonies were isolated and grown in liquid culture to express and secrete soluble nAbs. After centrifugation to remove bacteria, we validated the resultant nAb-containing bacteria culture supernatants (BC supes) by conventional ELISA against wells of microtiter plates coated with the respective purified recombinant target protein fragments. We next isolated phagemid DNA from the ELISA-positive bacterial colonies for sequencing. A summary of the results from these efforts is shown in *Table 1*. Plasmids with unique nAb sequences were archived as glycerol stocks and used as a source of DNA for cloning into mammalian expression plasmids for expression as intrabodies in mammalian cells or used as a source of bacterially-secreted nAbs for use as immunolabels.

## A subset of ELISA-positive nAbs function as intrabodies that recognize exogenously expressed target proteins in mammalian cells

We determined which unique nAbs could function as intrabodies when expressed in the reducing environment of the mammalian cell cytoplasm. cDNA inserts encoding each of the 113 unique ELISA-positive nAbs listed in *Table 1* were transferred from the pComb3XSS phagemid into the pEGFP-N1 or pEYFP-N1 mammalian expression plasmids by Gibson assembly (*Gibson et al., 2009*). This also involved the removal of the N-terminal bacterial leader sequence, addition of a start codon and removal of the sequences encoding the C-terminal PIII protein. The linker region containing 6XHis and HA tags between the nAb and C-terminal GFP was retained. After sequence verification, each of the 113 nAb mammalian expression plasmids was tested for nAb expression and intrabody function in mammalian cells by transient transfection into COS-1 cells. This assay entailed expression of each nAb and target protein either alone or together in separate wells of a 96 well microtiter plate. After two days of expression, cells were fixed, permeabilized, and subjected to IF-ICC performed with validated monoclonal antibodies (mAbs) against each target protein and analyses of the expression and subcellular localization of the nAb and target protein. Of note, COS-1 cells do not express detectable levels of any of these target proteins.

We found that the vast majority of COS-1 cells expressing EGFP- or EYFP-tagged nAb alone had substantial fluorescence signal in the nucleus (*Figure 1—figure supplement 1*). Substantial nuclear localization was also observed for these fluorescent proteins alone, as expected from prior studies showing nuclear localization of GFP [*e.g.,* (*Seibel et al., 2007*)]. In contrast, when the target proteins, such as Homer1 (*Figure 1—figure supplement 1*) were exogenously expressed in COS-1 cells, they were predominantly found in the cytoplasm, with the exception of the type I transmembrane protein

AMIGO-1 (*Kuja-Panula et al., 2003*), which primarily accumulates in the ER membrane (*Bishop et al., 2018*). We next evaluated whether target protein coexpression led to a change in nAb subcellular localization, which we interpreted as being due to nAb binding to the cytoplasmic- or ER-localized target protein. We visually determined whether coexpression of the predominantly nuclear nAb with the cytoplasmic/ER target protein would alter the nuclear localization of the nAb such that it colocalized with the target protein, as an indication that the nAb functioned as a target-binding intrabody. *Figure 1—figure supplement 1* shows a representative example of an anti-Homer1 nAb that was scored as a positive in this assay. While there was some variability in the extent of the target-protein-dependent impact on nAb localization between different cells in the population, and between different nAbs, likely reflecting cell-specific differences in the relative expression levels of nAb and target protein, the target protein-dependent altered distribution of nAbs provided a facile assay that allowed us to test every ELISA-positive nAb for whether they functioned as intrabodies in mammalian cells. A summary of the results of these intrabody screening assays on heterologous COS-1 cells is shown in *Table 1*.

## A subset of the nAbs that function as intrabodies in heterologous cells recognize endogenously expressed target proteins in cultured rat and mouse hippocampal neurons

These experiments in heterologous cells yielded a subset of nAbs that exhibited altered localization in the presence of target proteins when expressed in mammalian cells, indicating that they functioned as intrabodies. We next tested whether these nAbs could recognize endogenous target proteins when expressed in cultured rat hippocampal neurons (CHNs). Each of the GFP-or YFP-tagged nAbs that were scored as positive as intrabodies in heterologous cells was subsequently transfected into CHNs at 7–10 DIV. At 48 hr post-transfection, the neurons were fixed and subjected to IF-ICC with target-specific mAbs, or with markers for specific subcellular compartments. We evaluated colocalization between the GFP- or YFP-tagged nAbs and target protein immunolabeling, relative to CHNs expressing GFP or YFP alone. For each target, we identified nAbs that colocalized with the endogenously expressed target protein (results summarized in *Table 1*). Examples of CHNs expressing nAbs for each target with the corresponding target-specific immunolabeling are shown in *Figure 1—figure supplement 2* for the excitatory synaptic proteins Homer1, IRSp53 and SAPAP2, and *Figure 1—figure supplement 3* for Gephyrin and AMIGO-1. Analysis by Pearson's Correlation Coefficient (PCC) demonstrated that the nAb-GFP intrabodies were colocalized with the corresponding endogenous target protein (*Figure 1—figure supplements 2* and *3*). Importantly, the expression and subcellular localization of the endogenous target proteins were not altered by nAb expression, as quantified by a lack of significant change in target protein puncta size (*Figure 1—figure supplements 2* and *3*). For nAbs targeting the excitatory postsynaptic target proteins Homer1, IRSp53 and SAPAP proteins (*Figure 1—figure supplement 2*), the punctate pattern of nAb localization in dendrites was closely associated with immunolabeling for PSD95, a marker of the excitatory postsynaptic compartment (*Figure 1—figure supplement 4*), supporting that the nAb puncta were at excitatory synapses. Moreover, expression of the nAbs against target proteins at excitatory synapses did not impact the sizes of PSD-95 puncta (*Figure 1—figure supplement 4*). Similarly, anti-Gephyrin nAbs had a subcellular localization that not only colocalized with endogenous Gephyrin (*Figure 1—figure supplement 3*) but were also frequently found opposed to immunolabeling for the synaptic vesicle protein Synapsin (*Figure 1—figure supplement 5*), supporting the observation that these nAbs were localized at synapses. The sizes of the puncta of Gephyrin and Synapsin immunolabeling were not impacted by expression of anti-Gephyrin nAbs (*Figure 1—figure supplements 3* and *5*). In the case of AMIGO-1, we identified nAbs that colocalized with endogenous AMIGO-1 (*Figure 1—figure supplement 3*), which is present in large clusters at ER-PM junctions on the soma and proximal dendrites of CHNs (*Bishop et al., 2018*). Moreover, the AMIGO-1 nAb colocalized with labeling for Kv2.1, a partner subunit of AMIGO-1 in Kv2 channel complexes found at these sites (*Figure 1—figure supplement 5*). The sizes of AMIGO-1 and Kv2.1 puncta were not impacted by expression of anti-AMIGO-1 nAbs (*Figure 1—figure supplements 3* and *5*). These results indicate that these nAbs act as intrabodies that recognize their endogenous target proteins in CHNs, and the expression of the nAbs has little discernible impact on the expression and subcellular localization of their endogenous targets.

A subset of the anti-Homer1 nAbs were expressed in mouse CHNs over a longer time period from recombinant lentivirus under the control of the neuron-enriched Synapsin promoter (*Figure 2A*). At 10 days post-infection CHNs were subjected to IF-ICC with an anti-Homer1 mAb and an antibody against the dendritic marker MAP2. As shown in *Figure 2B*, in spite of such long-term expression of the cytoplasmically synthesized nAbs, nAbs such as HC20 exhibited little cytoplasmic accumulation of the nAb-YFP fluorescence (green) within the cell body (asterisk in the merged panel in *Figure 2B*). However, there was robust nAb-YFP fluorescence throughout the MAP2-positive (blue) dendrites of the expressing neuron, where it was present in puncta that precisely overlapped with punctate anti-Homer1 immunolabeling (red) (*Figure 2B*). An analysis of the localization of a number of anti-Homer1 nAb-YFP fusions after such long-term expression revealed that a subset of the nAb-YFP fusions (green), such as HC20 and HC87, exhibited precise colocalization with Homer1 immunolabeling (red), with little or no detectable signal in dendritic shafts, consistent with their binding to and accumulation at Homer1 clustered in dendritic spines (*Figure 2C,D*). However, other nAbs yielded more diffuse localization throughout the dendritic shafts, similar to that obtained with YFP alone, although most of these nAbs also exhibited a significantly higher degree of colocalization with Homer1 than YFP alone (*Figure 2C,D*). In spite of these differences in binding, in no case did long-term expression of the anti-Homer1 nAbs lead to significant changes in the overall sizes or density of Homer1 puncta (*Figure 2D*), showing that the long-term expression of these nAbs and their binding to synaptic Homer1 did not detectably impact Homer1 expression and localization.

Compared to similarly sized mobile transport vesicles, synaptic puncta are relatively stable structures. We next determined whether the discrete target protein-containing puncta labeled by GFP-tagged nAbs expressed as intrabodies in CHNs label stable structures or target protein containing transport vesicles (or other mobile structures). We performed short term time-lapse total internal reflection fluorescence (TIRF) imaging of live nAb-expressing CHNs. Over the course of one-minute recordings collected at 1 Hz, we observed that the nAb-GFP puncta (which we observed colocalized with the target proteins in the experiments mentioned in the previous section) were marking stable structures present in the TIRF field, such that overlaid time-lapse images of nAb-GFP localization yielded tightly colocalized structures (*Figure 3*). Similar to what we observed in fixed neurons (*Figure 2*; *Figure 1—figure supplements 2* and *3*), in live neurons GFP expression was diffuse and did not form punctate structures (*Figure 3*). These live cell imaging experiments are consistent with the immunocytochemistry date above that nAbs expressed in neurons accumulate at subcellular sites of stable target protein clustering (synapses, ER-PM junctions) and provide further evidence that these nAbs act as intrabodies to stably bind to their targets in living CHNs.

## A distinct subset of nAbs function as immunolabels in heterologous cells and brain sections

Because of their relatively small size (≈15 kD, ≈2–4 nm in length), nAbs are potentially advantageous for immunolabeling due to enhanced sample penetration and a reduced distance between the labeling signal and target compared to conventional antibodies. However, a systematic evaluation of the utility of nAbs for immunolabeling endogenous target proteins in brain neurons has not been reported. Here, we took an unbiased approach by testing all of the ELISA-positive nAbs for immunolabeling of conventional formaldehyde fixed transfected heterologous cells and brain sections. We first transformed *E. coli* with each of the ELISA-positive nAb phagemids, and after induction of nAb expression and overnight culture, collected the bacterial cell culture media as bacterial culture supernatants (BC supes) containing secreted nAbs. We first tested these nAb-containing BC supes in IF-ICC against fixed COS-1 cells transiently transfected to heterologously express their cognate brain target protein in a subset of cells, an assay we routinely use for screening mAbs (*Bekele-Arcuri et al., 1996*; *Gong et al., 2016*). This assay allows for a facile determination of which candidate antibodies detect the target protein after fixation by providing a mosaic of numerous target-expressing and non-expressing cells in the same field. Here, we double-immunolabeled cells with candidate nAb BC supes and previously validated mouse mAbs against the same target protein to distinguish expressing versus non-expressing cells. We found that a substantial subset (33/39) of the ELISA-positive anti-Homer1 nAb BC supes tested exhibited robust immunolabeling of fixed and permeabilized COS-1 cells expressing Homer1L. Examples of positive immunolabeling with anti-Homer1 nAb BC supes are shown in *Figure 1—figure supplement 6*, which shows robust nAb labeling

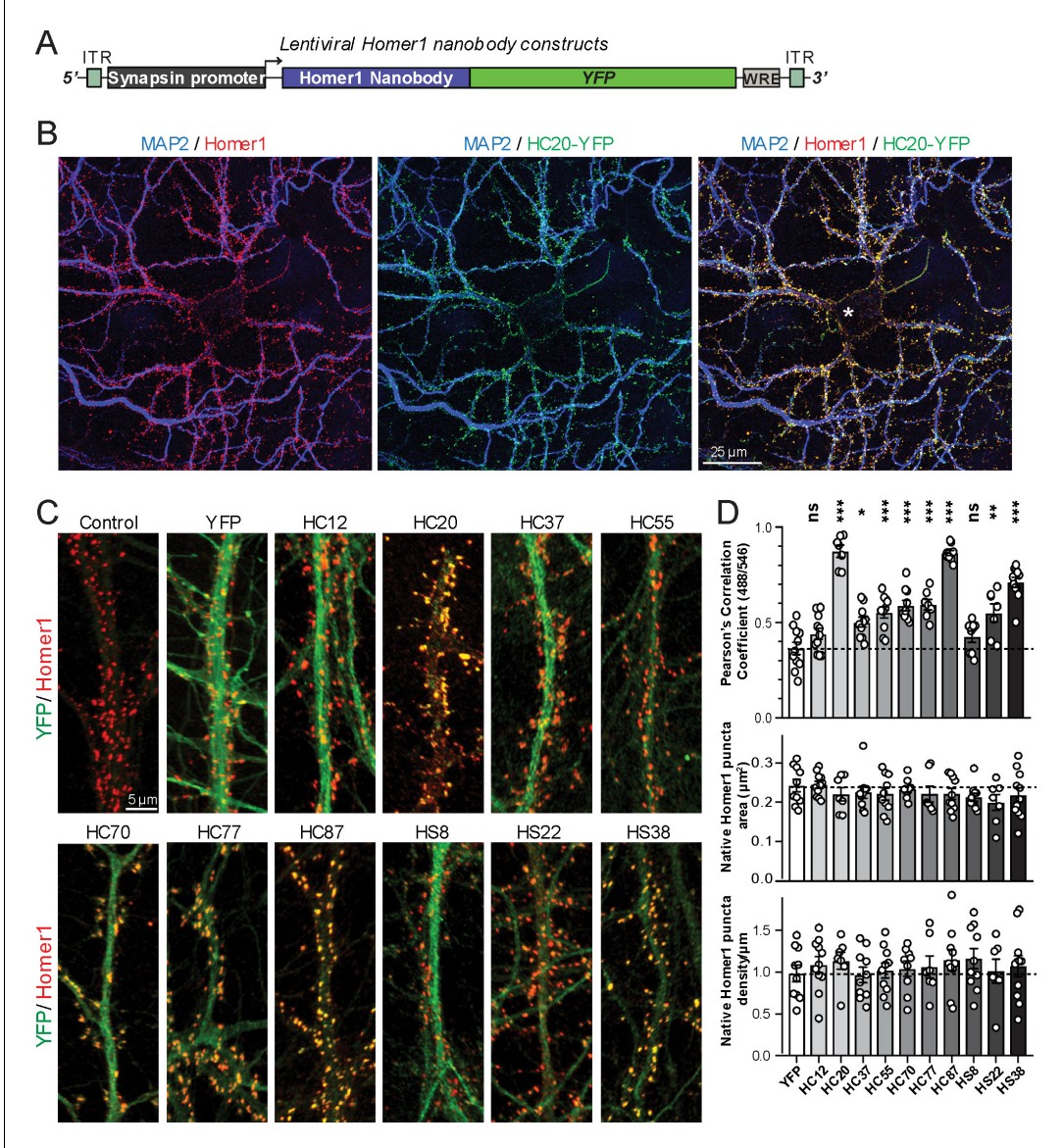

**Figure 2.** Anti-Homer nAbs expressed as intrabodies from recombinant lentivirus target to dendritic spines in cultured hippocampal neurons. (**A**) Schematic of the lentivirus targeting construct. (**B**) Image shows a representative field of a CHN culture infected at 5 days in vitro (DIV) with recombinant lentivirus encoding the anti-Homer1 HC20-YFP nAb fusion (green) and imaged at 14 DIV after immunolabeling for endogenous Homer1 (red) and the dendritic marker MAP2 (blue). (**C**) Images show representative fields of dendrites of infected CHNs expressing different nAb-YFP fusions as indicated (green) and immunolabeled for endogenous Homer1 (red). The scale bar in the top left Control panel is 5 μm and holds for all panels in C. (**D**) The top graph shows Pearson's Correlation Coefficient values between YFP or the different nAb-YFP fusions and anti-Homer1 immunolabeling. *p<0.01; **p<0.001; ***p<0.0001 for values of different anti-Homer1 nAb-YFP fusions versus for YFP alone. ns = not significant versus YFP alone. Values were analyzed by a one-way ANOVA followed by a Dunnett's post hoc test. The middle graph shows a size analysis of anti-Homer1 Ab labeled synaptic puncta in CHNs expressing YFP or the different anti-Homer1 nAb-YFP fusions. The bottom graph shows the density of anti-Homer1 Ab labeled synaptic puncta in CHNs expressing YFP or the different anti-Homer1 nAb-YFP fusions. Values for the size and density of anti-Homer1 Ab labeled synaptic puncta in CHNs expressing different anti-Homer1 nAb-YFP fusions are not significantly different than in CHNs expressing YFP alone. Bars on all graphs are mean ± S.E.M.

DOI: https://doi.org/10.7554/eLife.48750.010

(green) that colocalizes with Homer1 mAb labeling (red). In contrast, none of the unique ELISA-positive nAb BC supes against the other targets yielded detectable immunoreactivity in this assay (**Table 1**).

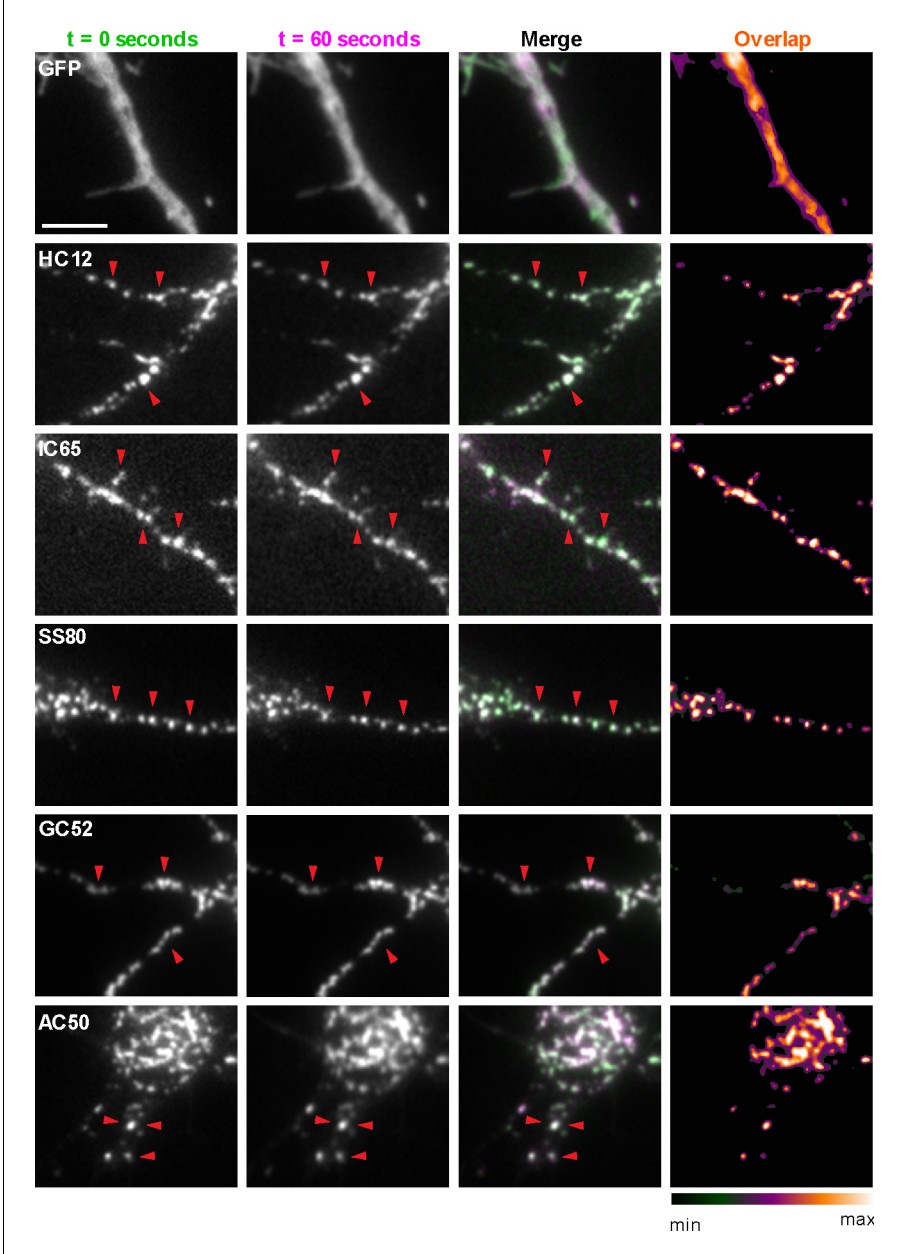

**Figure 3.** nAbs that function as intrabodies localize to immobile structures in cultured rat hippocampal neurons. TIRF images of live cultured rat hippocampal neurons transfected with GFP (shown in A), or nAb-GFP fusions against Homer1 (clone HC12, shown in B), IRSp53 (clone IC65, shown in C), SAPAP2 (clone SS80, shown in D), Gephyrin (clone GC52, shown in E), or AMIGO-1 (clone AC50, shown in F). For each, two images (of the same field of view) taken one min apart are shown. To the right is an overlay of the initial image (in green) and the subsequent image (in magenta). Overlap of green and magenta yields a white signal. Arrows point to punctate structures. The column to the far right shows an analysis of the extent of overlap of pixels between the initial and subsequent images. The scale bar in the top left panel is 5 µm and holds for all panels in figure.
DOI: https://doi.org/10.7554/eLife.48750.011

We next determined whether the anti-Homer1 nAbs exhibiting positive immunolabeling of Homer1L exogenously expressed in COS-1 cells would label endogenous Homer1 in brain sections. We evaluated each of the BC supes for labeling in sections prepared from perfusion-fixed adult rat brains. We found that a large proportion (25/33) of the BC supes tested yielded robust immunolabeling of brain sections, with the remainder exhibiting no detectable immunolabeling (*Table 1*). Each

of the 25 brain IHC-positive nAbs was also scored as positive in the heterologous COS-1 cell IF-ICC assay. As shown in *Figure 4A* for nAb HS15, the positive anti-Homer1 nAbs exhibited substantial immunolabeling of neuropil in caudate putamen (CPU), cerebral cortex (CTX) and hippocampus (HC). In hippocampus, immunolabeling for the positive anti-Homer1 nAbs (note that the HS12 nAb was scored as negative in this application) was especially prominent in subiculum and CA1 stratum oriens and radiatum (*Figure 4B*). The pattern of immunolabeling obtained with the anti-Homer1 nAbs matched closely that obtained with validated anti-Homer1 mouse mAbs (*Figure 4A,B*). While Homer1 labeling in the CA1 region of the hippocampus was consistently high for all nAbs that were scored as exhibiting positive immunolabeling, different nAbs exhibited variation in signal intensity in the CA2-CA3 regions and dentate gyrus (*Figure 4*). All nAbs that showed positive immunolabeling in these areas also had strong signals in olfactory bulb, and relatively weak or undetectable labeling in numerous other brain regions (thalamus, globus pallidus, brainstem, cerebellum), consistent with Homer1 mRNA expression in brain (*Clifton et al., 2017*; *Lein et al., 2007*). Furthermore, the nAb labeling pattern matched the localization of Homer1 protein from previous IHC studies (*Shiraishi et al., 2004*; *Shiraishi-Yamaguchi and Furuichi, 2007*), and immunolabeling obtained with anti-Homer1 mAbs recognizing both the long and short (L113/130) or only the long (L113/27) splice variants (*Figure 4C*). These results support that these nAbs can specifically bind to Homer1 in brain tissue.

## Directly conjugated nAbs function as nanoscale immunolabeling reagents for super-resolution fluorescence imaging of subcellular structures

To fully utilize the small size of nAbs as nanoscale immunolabeling reagents requires direct conjugation to detection reagents such as fluorescent organic dyes or gold particles. We selected a subset of nAbs that immunolabeled brain sections and made purified nAb preparations, taking advantage of the 6XHis tag engineered into the nAb C-terminus. We first validated the purified nAbs by immunolabeling brain sections (data not shown). We next directly conjugated a purified nAb (HS69) to the fluorescent organic dye Alexa Fluor 647 (Alexa647). We performed super-resolution microscopy employing a ground-state depletion system (*Bretschneider et al., 2007*; *Fölling et al., 2008*) to determine whether the Alexa647-nAb provided enhanced spatial resolution. In these experiments we compared CHNs immunolabeled with the directly conjugated HS69 nAb to that obtained with unlabeled HS69 nAb detected with Alexa647-conjugated anti-HA mAb, and that from a conventional anti-Homer1 mouse mAb (L113/27) detected with Alexa647-conjugated secondary antibody. As shown in *Figure 5*, there was a significant difference in the size of immunolabeled puncta for each of the labeling reagent combinations used. Directly conjugated Alexa647 nAb had the smallest overall cluster size, followed by the unlabeled nAb detected with the Alexa647-conjugated anti-HA antibody, with the conventional mAb detected with an Alexa647-conjugated secondary antibody having the largest cluster size (*Figure 5*). This provides an empirical demonstration that immunolabeling with a directly labeled nAb allows for enhanced resolution of target detection in super-resolution light microscopy.

## Validation of nAbs as immunolabels for use in immunoblot analyses

To complete the validation of nAbs for standard immunolabeling applications, we evaluated the utility of nAbs in our collection as immunolabels for detecting their respective target proteins in brain samples on immunoblots. We evaluated all 113 ELISA-positive unique nAb BC supes for immunolabeling of 'strip' blots containing a crude fraction of rat brain membrane proteins (*Bekele-Arcuri et al., 1996*; *Gong et al., 2016*; *Rhodes et al., 1995*). We found that a subset of the unique ELISA-positive nAbs against Homer1 (13/39; *Table 1*) detected a single robust band at the characteristic electrophoretic mobility of Homer1 ($\approx 47$ kDa) (*Saito et al., 2002*). The band recognized by the nAbs had characteristics corresponding to the band obtained with the validated anti-Homer1 mouse mAb L113/130 (*Figure 6*). A small subset of the unique ELISA-positive nAbs against Gephyrin (2/24; *Table 1*) detected a single band at the characteristic electrophoretic mobility ($\approx 80$ kDa) of Gephyrin (*Feng et al., 1998*) that comigrated with the immunoreactive band for the validated anti-Gephyrin mouse mAb L106/93 (*Figure 6*). No specific immunolabeling was detected with any of the

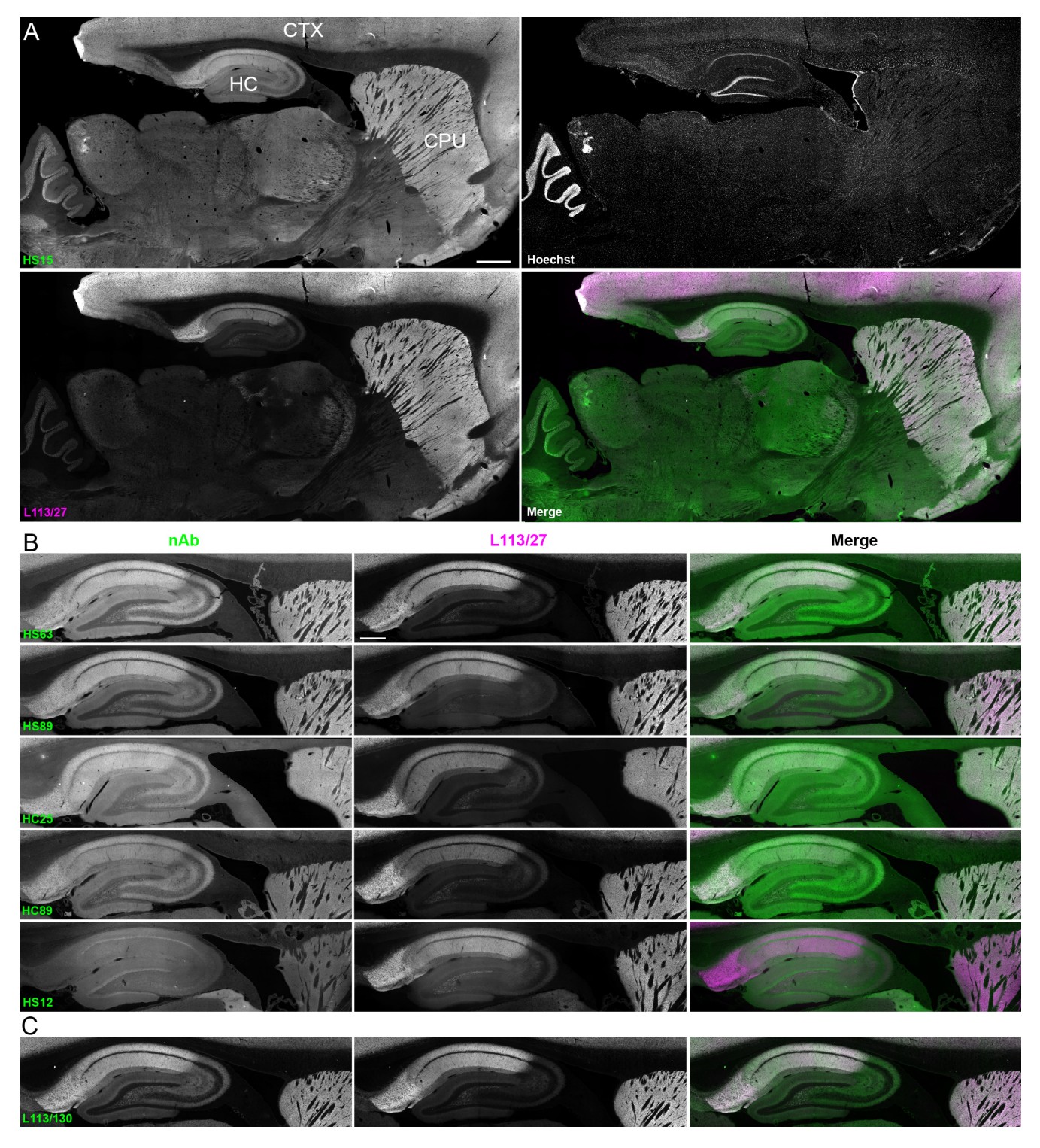

**Figure 4.** nAbs against Homer1 label rat brain sections through fluorescent immunohistochemistry. (**A**) Representative images of a whole brain section labeled with a nAb BC supe against Homer1 (HS15, in green), a mouse mAb against Homer1 (L113/27, in magenta) and a nuclear label (Hoechst, in grayscale) showing brain regions with high Homer1 labeling. Merge is of antibody labeling only. CPu = caudate putamen; CTX = cerebral cortex; HC = hippocampus. The scale bar in the HS15 panel is 1 mm and holds for all four panels. (**B**) Representative images of HC and CPu from brain sections labeled with several anti-Homer1 nAb BC supes (green) and a mouse mAb against Homer1 (L113/27, in magenta). Note that nAb HS12 was scored as negative in this application. The scale bar in the top L113/27 panel is 500 µm and holds for all panels in B and C. (**C**) Representative images

*Figure 4 continued on next page*

*Figure 4 continued*

from brain sections labeled with mouse mAb L113/130 that recognizes both the long and short splice variants of Homer1 (green), and mouse mAb L113/27 that recognizes only the long splice variants of Homer1 (magenta).

DOI: https://doi.org/10.7554/eLife.48750.012

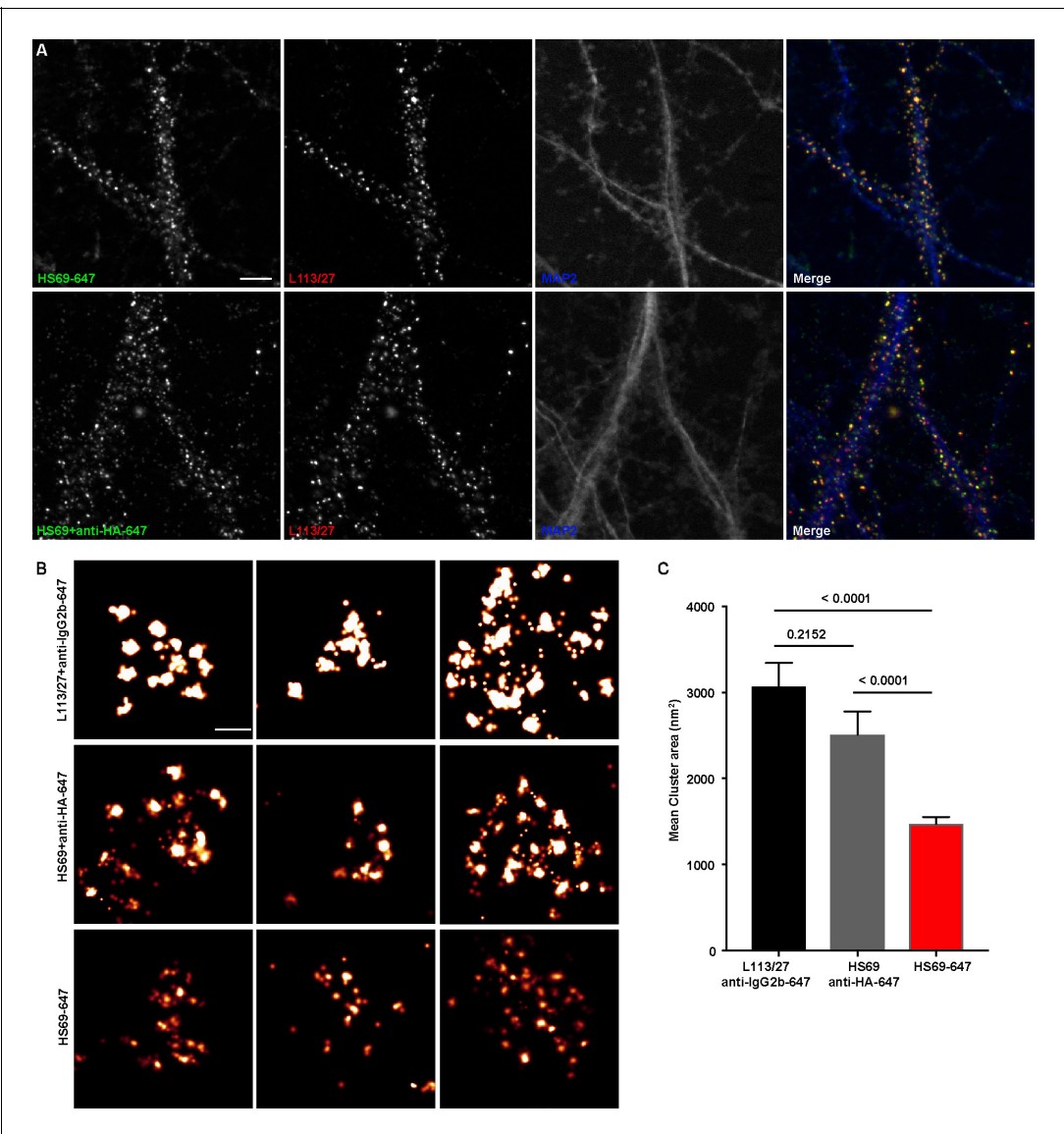

**Figure 5.** Immunolabeling with nAbs enhances spatial resolution. (**A**) Representative TIRF images of CHNs. Neurons were immunolabeled with anti-Homer1 nAb plus a mouse mAb against Homer1 (L113/27, in red), and an antibody against MAP2 (blue). Top row: Dendrites of a CHN immunolabeled with nAb HS69 directly conjugated to Alexa647 (HS69-647, in green). Bottom row: Dendrites of a CHN immunolabeled with HA-tagged HS69 nAb plus Alexa647 conjugated anti-HA mAb (HS69 + anti-HA-647, in green). Scale bar is 5 µm and holds for all panels in A. (**B**) Representative super-resolution Homer1 localization maps of neurons immunolabeled for Homer1 with a mouse mAb plus Alexa647-conjugated secondary Ab (L113/27 + anti IgG2a-647), with HS69 nAb plus Alexa647-conjugated anti-HA antibody (HS69 + anti-HA-647), or with directly Alexa647-conjugated HS69 nAb (HS69-647). Scale bar is 200 nm and holds for all images in panel B. (**C**) Bar plot of the mean Homer1 cluster area ± SD for the three different labeling groups described in B (n = 227, 403, and 593 clusters respectively from five different cells per group). *p* values were calculated using a one-way ANOVA and a Bonferroni's multiple comparisons test.

DOI: https://doi.org/10.7554/eLife.48750.013

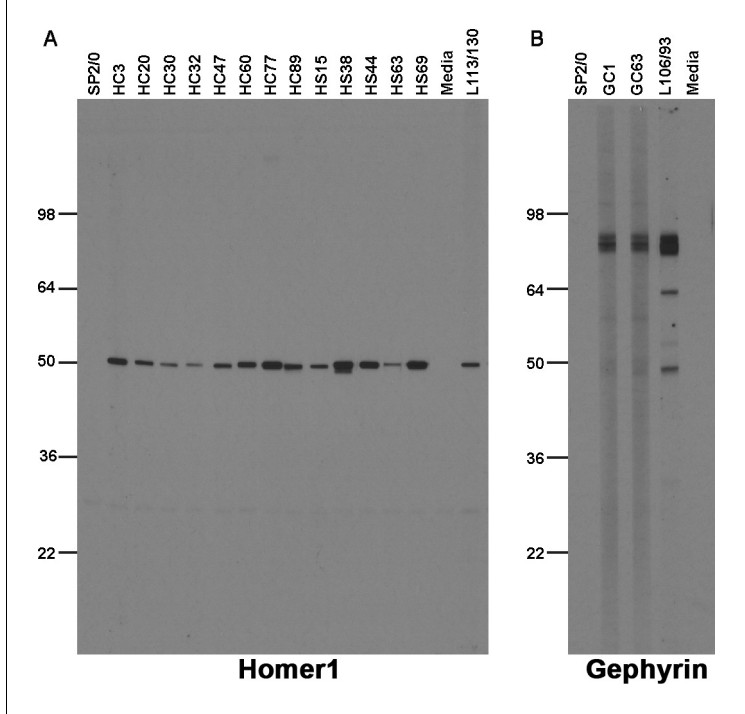

**Figure 6.** Immunolabeling with nAbs on immunoblots against a crude rat brain membrane fraction. (**A**) Positive immunoblot labeling with anti-Homer1 nAbs expressed from *E. coli* Top10F' cells; (**B**) Positive immunoblot labeling with anti-Gephyrin nAbs expressed from *E. coli* Top10F' cells. Control lanes are the respective positive control monoclonal antibodies, and negative controls the SB bacterial culture medium ('media') and conditioned medium from culture of the non-secreting SP2/0 myeloma cell line ("SP2/0").
DOI: https://doi.org/10.7554/eLife.48750.014

ELISA-positive nAb BC supes against the other targets (*Table 1*). These results demonstrate that a subset of ELISA-positive nAbs can be used in IB-based applications.

## Discussion

In this study, we report the development and validation of a novel toolbox of nAbs targeting a select set of neuronal proteins with specific subcellular localizations at sites mediating crucial neuronal signaling events. These include the postsynaptic scaffolding proteins Homer1 (*Brandstätter et al., 2004*), IRSp53 (*Soltau et al., 2002*), and SAPAP2 (*Takeuchi et al., 1997*), present at partially overlapping sets of excitatory synapses, Gephyrin (*Kneussel et al., 2001*), found at most inhibitory synapses, and the Kv2 channel auxiliary subunit AMIGO-1, found at ER-PM junctions on the soma, proximal dendrites and AIS (*Bishop et al., 2018*). We validated these nAbs for use as intrabodies in living neurons, and a subset as immunolabels in ICC, IHC, and IB applications. Since their initial discovery, nAbs have been employed in diverse applications in biomedical research (*Hamers-Casterman et al., 1993*; *Muyldermans, 2013*; *Desmyter et al., 2015*; *Beghein and Gettemans, 2017*; *Könning et al., 2017*; *De Meyer et al., 2014*) and have shown promise as human therapeutics (*Könning et al., 2017*; *Steeland et al., 2016*). However, few nAbs have been developed against neuronal proteins for use in neuroscience research applications. We report the feasibility of generating nAbs against a diverse set of neuronal target proteins with restricted subcellular distribution at sites mediating crucial neuronal signaling events and their validation in brain neurons as intrabodies and immunolabels, representing a valuable toolbox for neuroscience research applications that will be made publicly available in plasmid form.

Our approach relied on isolating nAbs from a library derived from immunized llamas. Numerous other studies demonstrated the feasibility of isolating nAbs from synthetic libraries, or libraries derived from nonimmunized animals (*de Kruif et al., 1995*; *Liang et al., 2007*; *Park et al., 2005*).

However, these approaches often yield low affinity binders that need to be matured by molecular evolution in vitro (*Gustchina et al., 2009*; *Gram et al., 1992*; *Jackson et al., 1995*), which is a time and labor-consuming process. Therefore, we elected to use immunizations to increase the representation of high affinity binders in the llama used as the source of our nAb cDNA library. One advantage of using large animals, like llamas, is that each animal can be simultaneously immunized with multiple targets (*Pardon et al., 2014*). This reduces the number of animals used and has the advantage that obtaining leukocytes for cDNA library generation does not require that the animal is sacrificed.

Our overall strategy generated at least one nAb that functioned as an intrabody in neurons for each of the five targeted brain proteins. These nAbs have potential use in a variety of research applications in living neurons, including visualizing endogenous target protein localization and dynamics, or modifying expression or function of the target protein. Moreover, they can be used to deliver cargo to specific sites, for example genetically-encoded $Ca^{2+}$ or membrane potential indicators to report on signaling events occurring in the specific subcellular compartments in which the target proteins are selectively localized. In addition, they could be used to target actuators to specific sites to locally modify membrane potential, enzymatic activity, or specific cell signaling events. The nAb intrabodies that we developed provide an array of specificities to target discrete neuronal compartments (excitatory and inhibitory synapses, ER-PM junctions) that play crucial and distinct roles in neuronal function, and importantly, do so without detectably altering the expression and localization of their target proteins. That the nAbs we evaluated by live cell imaging label stable structures within cultured rat hippocampal neurons suggests that the nAbs are primarily binding their targets at final subcellular location, as opposed to binding to the proteins while in transit to these sites, or otherwise targeting mobile structures. This may contribute to the lack of a detectable effect of nAb expression on target protein expression and localization.

In addition to developing nAbs as intrabodies for use in live cells, we also validated a subset of anti-Homer1 nAbs for use as immunolabels that recognize Homer1 in aldehyde-fixed samples of brain neurons. These nAbs exhibit the enhanced resolution in super-resolution light microscopy predicted for such nanoscale labeling reagents (*Beghein and Gettemans, 2017*; *Ries et al., 2012*; *Szymborska et al., 2013*; *Pleiner et al., 2018*). It is surprising to us that among the five proteins targeted in the nAb development efforts described here, only the Homer1 nAb project yielded large pools of nAbs positive for immunolabeling transfected heterologous cells and brain sections (*Table 1*), given that we had previously performed mouse mAb projects against each of these targets, using the same recombinant proteins as immunogens and in screening, and for each obtained a substantial number of mAbs positive for these applications (*Supplementary file 1*). We do not know the basis of this distinction between the Homer1 nAb project relative to the other nAb projects, and the overall differences between the corresponding llama nAb and mouse mAb projects. However, a retrospective analysis of the antiserum from the immunized llama by IF-ICC against transfected COS-1 cells expressing the individual target proteins yielded substantial immunolabeling only for cells expressing Homer1 (not shown), suggesting a fundamental lack of an immune response against epitopes preserved in aldehyde-fixed samples, and not a failure of phage display selection to capture immunolabeling-competent nAbs. A similar scenario held for epitopes present on SDS-denatured target proteins detected on immunoblots blots, with the Homer1 nAb project yielding substantial numbers of positive nAbs, the Gephyrin project two, and the other projects none (*Table 1*), again in contrast to the corresponding mouse mAb projects (*Supplementary file 1*). It has been suggested that nAbs preferentially bind to target proteins via a convex paratope (*Wurch et al., 2012*), which could contribute to the inability of the nAbs against the targets with the exception of Homer1 to detect aldehyde-fixed and/or denatured target proteins. While nAbs have widespread use in binding to native protein, which enhances their utility as chaperones for crystallography (*Pardon et al., 2014*), certain nAbs generated against GFP and other proteins have been used as immunolabels for immunohistochemistry (*Fang et al., 2018*; *Yamagata and Sanes, 2018*) and immunoblots (*Bruce and McNaughton, 2017*).

In summary, we have generated a series of validated nAbs against neuronal proteins selectively expressed in specific subcellular compartments in brain neurons. These nAbs represent a valuable toolbox of reagents available to the neuroscience community for diverse applications. The pipeline we employed is an exemplar that can be used in developing nAbs against other neuronal targets to enhance the spectrum of experimental approaches available to researchers.

# Materials and methods

**Key resources table**

| Reagent type (species) or resource | Designation | Source or reference | Identifiers | Additional information |
|---|---|---|---|---|
| Cell line (*Cercopithecus aethiops*) | COS-1 | ATCC Cat # CRL-1650, Lot # 59102713; PMID: 6260373 | RRID:CVCL_0223 | |
| Antibody | numerous | | | See *Supplementary file 4* |
| Recombinant DNA reagent | pComb3XSS | PMID: 10986398 | Addgene #63890 (Addgene RRID:SCR_002037) | |
| Software algorithm | Photoshop | Adobe Systems | RRID:SCR_014199 | |
| Software algorithm | Axiovision | Carl Zeiss MicroImaging | RRID:SCR_002677 | |
| Software algorithm | Fiji | PMID: 22743772 | RRID:SCR_002285 | |

## Animals

All procedures involving llamas were performed at Triple J Farms of Kent Laboratories (Bellingham, WA) in strict accordance with the Guide for the Care and Use of Laboratory Animals of the NIH. All procedures involving rats were approved by the University of California, Davis, Institutional Animal Care and Use Committee (IACUC) under protocols 20485 and 21265 and were performed in strict accordance with the Guide for the Care and Use of Laboratory Animals of the NIH. All rats were maintained under standard light-dark cycles and allowed to feed and drink ad libitum. All procedures involving mice were approved by the Stanford University IACUC under protocol 18846 and were performed in strict accordance with the Guide for the Care and Use of Laboratory Animals of the NIH.

## Llama immunization and characterization of immune responses

Recombinant fragments of neuronal target proteins (*Supplementary file 2*) were expressed in and purified from *E. coli*. Llama immunizations were performed using cocktail of these recombinant protein fragments employing five subcutaneous injections given at biweekly intervals. For each injection a total of 1 mg protein (200 µg of each protein in the cocktail) was mixed with Freund's complete adjuvant (Sigma-Aldrich Cat# F5581) for the first immunization and Freund's incomplete adjuvant (Sigma-Aldrich Cat# F5506) for all subsequent immunizations. The llama immune response against the individual target proteins was evaluated by ELISA beginning with antiserum collected after the third immunization.

A whole IgG fraction was purified from llama antiserum on rProtein A/G GraviTrap columns (GE Healthcare Cat# GE28-9852-56). Briefly, 0.5 mL of llama antiserum was diluted with an equal volume of phosphate-buffered saline (PBS, 5.2 mM $Na_2HPO_4$; 1.7 mM $KH_2PO_4$; 0.15 M NaCl, pH 7.4). The mixture was loaded onto a Protein A/G GraviTrap column, the unbound fraction collected, and the column washed with 20 mL PBS. The whole IgG fraction was eluted with 0.1 M Glycine-HCl buffer (pH 2.7). All fractions were neutralized with 1:10 vol of 1 M Tris-HCl (pH 9.0) upon collection.

Conventional heavy and light chain IgG1 and heavy chain-only IgG2 and IgG3 subclasses were purified from llama antiserum on rProtein A GraviTrap (GE Healthcare Cat# GE28-9852-54) and Protein G GraviTrap (GE Healthcare Cat# GE28-9852-55) columns. Briefly, 0.5 mL of llama antiserum was diluted with an equal volume of phosphate-buffered saline (PBS, 5.2 mM $Na_2HPO_4$; 1.7 mM $KH_2PO_4$; 0.15 M NaCl, pH 7.4). The mixture was loaded onto a Protein G GraviTrap column, the unbound fraction collected, and the column washed with 20 mL PBS. The IgG3 fraction was first eluted with 58% acetic acid buffer (pH 3.5) containing 0.15 M NaCl. The IgG1 fraction was subsequently eluted with 0.1 M Glycine-HCl buffer (pH 2.7). The unbound fraction from the Protein G column was loaded onto on an rProtein A GraviTrap column, and the column washed with 20 mL PBS. The IgG2 fraction was eluted with 58% acetic acid buffer (pH 4.5) containing 0.15 M NaCl. All fractions were neutralized with 1:10 vol of 1 M Tris-HCl (pH 9.0) upon collection.

## Phage display library construction and panning

A phage display library for nAb isolation was prepared in the pComb3XSS phagemid (AddGene #63890). This allows for generation of 6xHis and HA-tagged nAbs as PIII protein fusions on phage pili for phage display, and also for subsequent production of soluble protein lacking the PIII fusion partner in non-amber suppressor *E. coli* strains (Andris-Widhopf et al., 2000). Llama blood was collected after the fifth immunization and 10 mL used for isolation of total RNA using a LeukoLOCK total RNA isolation system (ThermoFisher Cat# AM1923). The resulting total RNA was reverse-transcribed to cDNA using the SuperScript III First Strand Synthesis System (ThermoFisher Cat# 18080–051). The IgG heavy chain variable fragments were amplified by PCR with llama-optimized primers (**Supplementary file 3**). The amplified PCR product and the phagemid vector pComb3XSS were digested with the *Sfi* I restriction enzyme (NEB Cat# R0123S) and the resulting fragments ligated. The ligation was transformed by electroporation into electrocompetent ER2738 *E. coli* bacteria (Lucigen Cat# 60522–2). The transformants were infected with M13KO7 helper phage (NEB Cat# N0315S) for packaging into phage displaying the nAb library on pili as PIII fusions. The library complexity was calculated as $8.0 \times 10^7$.

Target-specific nAbs were enriched by panning against the individual target proteins bound to wells of 96 well microplates (Greiner Bio-One Cat# 655061). For conventional panning, microplate wells were coated overnight at 4°C with 10 µg of target protein in PBS. After blocking with BSA (5 mg/mL in PBS) for 1 hr at room temperature (RT, ≈22°C), $10^{11}$ phage particles were added to each well and incubated for 1 hr at RT. Following eight rinses with PBST buffer (0.5% Tween-20 in PBS) and eight rinses with PBS buffer to remove unbound phage, bound phage were eluted by incubation with 10 mg/mL trypsin (Sigma-Aldrich Cat# T4799) in TBSC (50 mM Tris–HCl pH 7.4, 100 mM NaCl, 1 mM $CaCl_2$) containing 3% (w/v) BSA for 30 min at 37°C.

The eluted phage were collected and used to reinfect 5 mL of log-phase ER2738 *E. coli* bacteria. After 2 hr of incubation, the reinfected culture was transferred to a 250 mL flask with the addition of 35 mL 2xYT medium, $10^{12}$ M13KO7 helper phage, and antibiotics (tetracycline: 20 µg/mL; kanamycin: 50 µg/mL; and ampicillin or AMP: 100 µg/mL) and the culture incubated overnight at 37°C with shaking at 250 rpm. Phage were collected and used for a second round of panning.

For Homer1, SAPAP2 and Gephyrin, a parallel round of panning was performed using target proteins displayed using a sandwich technique. After confirmation of immunoreactivity against each of the individual target proteins by ELISA, 10 µg of a whole IgG fraction purified from antiserum obtained from the immunized llama was added to individual wells of a 96 well microplate and incubated at 4°C overnight. After removing unbound IgG by two rinses with PBS, 10 µg of individual target protein was added to the separate wells and incubated for 1 hr at RT. Unbound target protein was removed by eight rinses with PBST and eight rinses with PBS. The remainder of the panning procedure was then carried out as described above.

## Production of soluble nAbs

Phage display positive samples were grown in liquid culture and plated onto LB + AMP plates. To prepare bacterially-expressed nAb culture supernatant (BC supe) for protein ELISA validation, isolated colonies were picked into 2 mL capacity deep well plates containing 0.5 mL of LB + AMP medium per well. After overnight incubation at 37°C with shaking at 250 rpm, 50 µL of these cultures were transferred into 1 mL of Super Broth + AMP until an $A_{600}$ of 0.3–0.6 was reached. Expression of the nAb-PIII fusion proteins was induced by addition of 1 mM isopropyl-β-D-thiogalactopyranoside and incubation for 16 hr at 37°C. The media was harvested, and the BC supe collected after centrifugation at 1990 x g for 15 min at RT to pellet the bacteria.

To prepare purified soluble nAbs lacking the PIII fusion protein, a subset of the nAb-pComb3XSS plasmids were isolated and transformed into the amber suppressor *E. coli* strain Top10F'. The induction of soluble nAb expression was performed in 1 L of Super Broth + AMP as described above. After pelleting the cells by centrifugation at 16,900 x g for 10 min at 4°C, the periplasmic proteins were released by osmotic shock (Olichon et al., 2007). The periplasmic extract was loaded onto a column of HisPur Ni-NTA Resin (ThermoFisher Cat# 88221), and after washing with 10 mL of PBS and 10 mL of 40 mM imidazole in PBS, the bound proteins were eluted with 5 mL of 200 mM imidazole in PBS. The eluted fraction was dialyzed against PBS to remove the imidazole. The protein

concentration was determined by $A_{280}$, and the concentration verified, and protein purity evaluated by SDS-PAGE and Coomassie blue staining.

## Phage and nAb ELISAs

The binding activity of nAbs in vitro was evaluated by ELISA. Individual wells of a 96 well microtiter plate (Greiner Bio-One Cat# 655081) were coated overnight at 4°C with 10 µg of the individual target proteins. After removing unbound target protein by four rinses with PBST buffer, wells were blocked for 1 hr at RT with BLOTTO (4% nonfat dry milk powder in TBS-T buffer). Phage (for phage ELISAs) or soluble nAb (for nAb ELISAs) samples were added, followed by incubation for 1 hr at RT. For phage ELISAs, $10^{10}$ phage particles were added per well, while for nAb ELISAs, 50 µL of BC supe was added per well. Following four rinses with PBST buffer, wells were incubated for 1 hr at RT with 50 µL of anti-HA mouse IgG2b mAb 12CA5 at 1.14 µg/mL in BLOTTO. Following four rinses with PBST, wells were incubated for 1 hr at RT with 50 µL of goat anti-mouse IgG2b-HRP conjugated secondary antibody (Jackson Immunoresearch Laboratories Cat# 115-005-207) diluted 1:5000 in BLOTTO. Following four rinses with PBST buffer, wells were incubated with TMB substrate (Sigma-Aldrich Cat# T2885). Wells lacking phage or nAbs were used as negative controls. Phage or nAbs exhibiting an $OD_{450}$ signal three-fold higher than the negative 'no primary' control were selected for further analysis. For target proteins that were fusions with GST (i.e., IRSp53 and AMIGO-1), parallel ELISAs were performed on plates whose wells were coated with an equal amount of GST protein. Phage or nAbs exhibiting a three-fold higher $OD_{450}$ signal for the GST-target protein fusion versus GST alone were selected for further analysis.

## Sequencing and transfer of unique nAbs into mammalian expression plasmids

DNA in pComb3XSS phagemid was prepared from the ER2738 strain of *E. coli* bacteria for all ELISA-positive nAbs and the forward strand of the nAb insert subjected was sequenced using a primer (5'-TTAGGCACCCCAGGCTTTACACT-3') that binds to the leader sequence of the pComb3XSS phagemid.

All unique ELISA-positive nAb sequences were amplified by PCR and ligated into the pEYFP-N1 or pEGFP-N1 mammalian expression plasmids as GFP fusions by Gibson Assembly, employing a commercial Master Mix (NEB Cat# E2611S) according to the manufacturer's protocol, and employing a SmaI (NEB Cat# R0141S) restriction site present in the pEYFP-N1 or pEGFP-N1 plasmid. The primers used for the Gibson assembly reaction were Forward: 5'-ATTCTGCAGTCGACGG TACCGCGGGCCCTGGTTTCGCTACCGTGGCCCAGGCGGCC-3' or 5'-CTTCGAATTCTGCAG TCGACGGTACCGCGGGGCCATGCAGKTGCAGCTCGTGGAGTC-3'; Reverse: 5'-CTCACCATGG TGGCGACCGGTGGATCCCTAGCGTAGTCCGGAACGTCGTACGGGTA-3'. Two independent colonies for each construct were selected for plasmid preparation, sequence determination and expression in mammalian cells. Plasmids encoding validated nAbs will be made publicly available in plasmid form from Addgene.

## Mammalian COS-1 cell culture and transfection

COS-1 cells were obtained from ATCC. These were verified by ATCC to be exclusively of *Cercopithecus aethiops* (African green monkey) origin by cytochrome oxidase I (COI) assay, and to be mycoplasma negative at the time of cryopreservation. COS-1 cell cultures were subsequently tested for mycoplasma contamination on a monthly basis using the MycoAlert Mycoplasma Detection Kit (Lonza Catalog#: LT07-318). COS-1 cells were maintained in Dulbecco's modified Eagle's medium (ThermoFisher Cat# 11995065) supplemented with 10% Bovine Calf Serum (HyClone Cat# SH30072.04), 1% penicillin/streptomycin (ThermoFisher Cat# 15140122), and 1X GlutaMAX (ThermoFisher Cat# 35050061) in a humidified incubator at 37°C with 5% $CO_2$. For testing nAbs for intrabody function, 2,500 COS-1 cells were plated in each well of a 96 well plate (Greiner Bio-One Cat# 655090), cultured overnight at 37 °C and then transfected using Lipofectamine 2000 (ThermoFisher Cat# 11668027) following the manufacturer's protocol. Sets of individual wells were transfected with nAb or target expression plasmids separately, or cotransfected together, using 0.1 µg each of each plasmid. Cells were transiently transfected in DMEM without supplements, then returned to regular growth media 4 hr after transfection. Cells were used 40–48 hr post-transfection for IF-ICC.

For testing nAbs for immunolabel function by IF-ICC, $3 \times 10^4$ COS-1 cells were plated on number 1.5 glass coverslips coated with poly-L lysine in 35 mm Petri dishes and incubated overnight. COS-1 cells were transiently transfected using Lipofectamine 2000 (ThermoFisher Cat# 11668027) with 1 µg of target protein plasmid following the manufacturer's protocol. Mammalian expression plasmids encoding full-length mouse Homer1 (Origene Cat# MR222523), human IRSp53 (Addgene Plasmid# 31656), rat SAPAP2 (Addgene Plasmid# 40216), human Gephyrin (a gift from Dr. Stephen J. Moss, Tufts University), and mouse AMIGO-1 (a gift from Dr. Heikki Rauvala, University of Helsinki) were used for expression of the corresponding target proteins in COS-1cells.

## Culture and transfection of rat hippocampal neurons

Hippocampi were dissected from embryonic day 18 rat embryos and dissociated enzymatically for 20 min at 37°C in 0.25% (w/v) trypsin (ThermoFisher Cat# 15050065) in HBSS and dissociated mechanically by triturating with glass polished Pasteur pipettes. Dissociated cells were suspended in plating medium containing Neurobasal (ThermoFisher Cat# 21103049) supplemented with 10% FBS (ThermoFisher Cat# 16140071), 2% B27 (ThermoFisher Cat# 17504044), 2% GlutaMAX (Thermo-Fisher Cat# 35050061), and 0.001% gentamycin (ThermoFisher Cat# 15710064) and plated at 60,000 cells per dish in glass bottom dishes (MatTek Cat# P35G-1.5–14 C) coated with 0.5 mg/mL poly-L-lysine (Sigma-Aldrich Cat# P2636). At 7 days in vitro (DIV), cytosine-D-arabinofuranoside (Sigma-Aldrich Cat# 251010) was added to a final concentration of 5 µM to inhibit non-neuronal cell growth. Neurons were transiently transfected with 1 µg of each nAb mammalian expression plasmid at 7–10 DIV using Lipofectamine 2000 (ThermoFisher Cat# 11668019) for 1.5 hr as described by the manufacturer. Neurons were used 40–48 hr post transfection for IF-ICC.

## Examination of Homer1 nAbs delivered via lentivirus

Primary hippocampal cultures were generated by dissecting hippocampi from P0 CD1 mice, and cells were dissociated by papain (Worthington Cat# LS003127) digestion for 20 min at 37°C, filtered through a 70 µm cell strainer (Falcon Cat# 352350), and plated on Matrigel (Corning Cat# 356235)-coated 0 thickness glass coverslips (Assistant Cat# 01105209) in 24-well plates. Plating media contained 5% fetal bovine serum (Atlanta Biologicals Cat# S11550), B27 (ThermoFisher Cat# 17504044), 0.4% glucose (Sigma Cat# G8270), 2 mM glutamine (ThermoFisher Cat# 25030164), in 1x MEM (ThermoFisher Cat# 51200038). Culture media was exchanged to Growth media 24 hr later (1 DIV), which contained 5% fetal bovine serum, B27, 2 mM glutamine in Neurobasal A (ThermoFisher Cat# 10888022). Cytosine arabinofuranoside (Santa Cruz Biotechnology Cat# 221454A) was added at a final concentration of 4 µM on or around 3 DIV based on glial cell density in a 50% growth media exchange. Cultures were subsequently infected at 5 DIV and analyzed at 14 DIV. For production of lentiviruses, the lentiviral expression shuttle vector and three helper plasmids [pRSV-REV, pMDLg/pRRE and vesicular stomatitis virus G protein (VSVG)] were co-transfected into HEK293T cells (ATCC), using 5 µg of each plasmid per 25 cm$^2$ culture area, respectively. Transfections were performed using the calcium-phosphate method in media lacking antibiotic (DMEM + 10% FBS). Media with viruses was collected at 48 hr after transfection, centrifuged at 5000 x g for 5 min to remove debris, and 50 µL of viral conditioned media was added directly to each well. For IF-ICC analysis, all solutions were made fresh and filtered with a 0.2 µm filter prior to starting experiments. Cells were washed briefly with PBS, fixed with 4% formaldehyde/FA (freshly prepared from paraformaldehyde/PFA)/4% sucrose/PBS for 20 min at 4°C, washed $3 \times 5$ min each in PBS, and permeabilized in 0.2% Triton X-100/PBS for 5 min at RT. Cells were subsequently placed in blocking buffer (5% BSA/PBS) for 1 hr at RT, and incubated with primary antibodies [*Supplementary file 4*; chicken anti-GFP (Aves Labs Cat# GFP1020, RRID:AB_10000240); rabbit anti-Homer1 (Synaptic Systems Cat# 160003, RRID: AB_887730), anti-MAP2 mouse IgG1 mAb AP-20 (Sigma Cat# M1406, RRID:AB_477171)] diluted in blocking buffer overnight at 4°C. Cells were washed $3 \times 5$ min each in PBS, incubated with diluted fluorescently-conjugated secondary antibodies (Life Technologies Cat# A11039, A11012, A21236) in blocking buffer for 1 hr, washed $3 \times 5$ min each in PBS, and mounted on UltraClear microscope slides (Denville Scientific Cat# M1021) using DAPI Fluoromount-G (Southern Biotech Cat# 010020).

## Immunofluorescence immunocytochemistry

Fixation of COS-1 cells was performed as previously described (*Bishop et al., 2015*). Briefly, COS-1 cells were fixed in ice-cold 4% FA (freshly prepared from PFA, Sigma-Aldrich Cat# 158127) in PBS containing 0.1% Triton-X 100 for 15 min at 4°C, and CHNs in ice-cold 4% FA/4% sucrose in PBS for 15 min at 4°C. All subsequent procedures were performed at RT. Cells were washed for 3 × 5 min each in PBS and blocked and permeabilized for 1 hr in BLOTTO + 0.1% Triton-X 100 (BLOTTO-T). For assays determining the efficacy of nAbs as intrabodies, incubation with primary antibodies (*Supplementary file 4*) was performed using mAb tissue culture supernatants (TC supes) diluted 1:2 in BLOTTO-T for 1 h T. Following primary antibody incubation, and 3 × 5 min each washes in BLOTTO-T, cells were incubated with mouse IgG subclass-specific Alexa Fluor-conjugated secondary antibodies (all secondary antibodies from ThermoFisher) diluted 1:1500 in BLOTTO-T containing 500 ng/mL Hoechst 33258 (ThermoFisher Cat# H1399) for 1 hr, washed 3 × 5 min each in PBS, and mounted onto microscope slides using Prolong Gold (ThermoFisher Cat# P36930). For assays determining the efficacy of nAbs as immunolabels, a 1 hr primary antibody incubation was performed using undiluted nAb BC supes, followed by a 1 hr incubation with mouse mAb TC supes (*Supplementary file 4*) diluted 1:5 in BLOTTO-T. Following these serial primary antibody incubations, and 3 × 5 min each washes in BLOTTO-T, cells were incubated with anti-HA mouse IgG1 mAb 16B12 conjugated to Alexa Fluor 488 (ThermoFisher Cat# A-21287) diluted to 0.67 µg/mL in blocking solution to detect bound nAb, and mouse IgG subclass-specific Alexa Fluor 555 conjugated secondary antibodies (all secondary antibodies from ThermoFisher) diluted 1:1500 in BLOTTO-T to detect bound mAb, with dilutions performed in BLOTTO-T containing 500 ng/mL of the chromatin dye Hoechst 33258 (ThermoFisher Cat# H1399). After a 1 hr incubation, cells were washed 3 × 5 min each in PBS and mounted onto microscope slides using Prolong Gold (ThermoFisher Cat# P36930).

## Immunofluorescence immunocytochemistry for super-resolution imaging

The HS69 anti-Homer1 nAb (100 µg in 1 mL of 0.1 M sodium bicarbonate buffer, pH 8.3) was directly conjugated to Alexa647 using succinimidyl-Alexa647 (ThermoFisher Cat# A20186) for 1 hr at RT, followed by addition of 0.1 mL of 1.5 M hydroxylamine. Conjugated nAb was separated from free dye on a Sephadex G-25 column, followed by dialysis against PBS. Validation of immunolabeling of CHNs at 24 DIV with directly conjugated HS69-647 nAb as shown in *Figure 5A* was performed essentially as described in the previous section. For primary and secondary antibody incubations, neurons were divided into two groups. CHNs in both groups were immunolabeled with primary antibodies mouse anti-Homer1 mouse IgG2b mAb L113/27 TC supe at a 1:2 dilution, and rabbit anti-MAP2 (Millipore-Sigma, Cat# AB5662-I) at a 1:100 dilution, and with secondary antibodies goat anti-mouse IgG2b-subclass-specific Alexa Fluor 555-conjugated secondary antibody (ThermoFisher Cat# A21147) at a 1:1500 dilution to detect the anti-Homer1 mouse IgG2b mAb L113/27, and goat anti-rabbit IgG (H+L) Alexa Fluor 350-conjugated secondary antibody (ThermoFisher Cat# A-11046) at 1:1500 dilution to detect the rabbit anti-MAP2. For one set of CHNs, the primary antibody cocktail also included anti-Homer1 nAb HS69 directly conjugated to Alexa 647 (HS69-647) diluted to 10 µg/mL. For the other set, the primary antibody cocktail included unconjugated anti-Homer1 HS69 nAb diluted to 10 µg/mL, and the secondary antibody cocktail anti-HA mouse IgG1 mAb 2–2.2.14 conjugated to Alexa 647 (ThermoFisher Cat# 26183-A647) diluted to 1 µg/mL.

For super-resolution imaging experiments (Figure 11B, C) CHNs at 24 DIV were fixed in ice-cold 3% FA (freshly prepared from PFA, Sigma-Aldrich Cat# 158127)/0.1% glutaraldehyde (Sigma-Aldrich Cat# G7651) in PBS for 15 min at 4°C. Unless otherwise stated, all remaining procedures were performed at RT. Fixed CHNs were washed for 3 × 5 min each in PBS, and free aldehydes reduced by incubating in 0.1% NBH$_4$ (Sigma-Aldrich Cat# 213462) in dH$_2$O for 5 min, followed by 3 × 5 min each washes in PBS. Neurons were blocked and permeabilized for 60 min in 3% BSA with 0.25% Triton-X 100 in PBS (blocking solution). For primary and secondary antibody incubations, neurons were divided into three groups. Group 1, mAb plus secondary: CHNs were incubated overnight at 4°C with anti-Homer1 mouse IgG2b mAb L113/27 TC supe at a 1:2 dilution in blocking solution, washed for 3 × 5 min each in blocking solution, and incubated for 1 hr with goat anti-mouse IgG2b secondary antibody conjugated to Alexa 647 (ThermoFisher Cat# A-21242) diluted to 1 µg/mL in blocking

solution, followed by 3 × 5 min each washes in PBS. Group 2, nAb plus anti-HA-647: CHNs were incubated overnight at 4°C with HS69 anti-Homer1 nAb diluted to 10 μg/mL in blocking solution, washed for 3 × 5 min each in blocking solution, and incubated for 1 hr with anti-HA mouse IgG1 mAb 2–2.2.14 conjugated to Alexa 647 (ThermoFisher Cat# 26183-A647) diluted to 1 μg/mL in blocking solution, followed by 3 × 5 min each washes in PBS. Group 3, directly-conjugated nAb: CHNs were incubated overnight at 4°C with anti-Homer1 nAb HS69 directly conjugated to Alexa 647 (HS69-647) diluted to 10 μg/mL in blocking solution followed by 3 × 5 min each washes in PBS. Coverslips were mounted on microscope slides with a round cavity (NeoLab Migge Laborbedarf-Vertriebs GmbH, Germany) and sealed with Twinsil (Picodent, Germany). The imaging buffer contained 10 mM Cysteamine-HCl (Sigma-Aldrich Cat# M6500), 0.56 mg/mL glucose oxidase (Sigma-Aldrich Cat# G2133), 3.4 μg/ml catalase (Sigma-Aldrich Cat# C100), and 10% w/v glucose in 200 mM Tris-HCl pH 8, 10 mM NaCl.

## Multiplexed fluorescence immunohistochemistry on brain sections

Rats were deeply anesthetized with 70 mg/kg Na-pentobarbital salt (Sigma-Aldrich Cat# P3761) in 0.9% NaCl solution through intraperitoneal injection, followed by boosts as needed. Once completely anesthetized, rats were transcardially perfused with 25 mL of ice-cold PBS containing 10 U/mL heparin, followed by an ice-cold fixative solution of 4% FA (freshly prepared from PFA, Sigma-Aldrich Cat# 158127) in 0.1 M sodium phosphate buffer (PB), pH 7.4, using a volume of 0.5 mL fixative solution per gram of rat weight. Following perfusions, brains were removed from the skull and cryoprotected in 10% sucrose in 0.1 M PB overnight at 4°C, then transferred to a solution of 30% sucrose in 0.1 M PB for 24–48 hr, until they sank to the bottom of the tube. Following cryoprotection, all brains were frozen, and cut on a freezing stage sliding microtome (Richard Allen Scientific) to obtain 30 μm-thick sagittal sections. Sections were collected in 0.1 M PB containing 10 mM sodium azide and processed for immunohistochemistry as free-floating sections.

Multiplex immunofluorescence labeling of nAbs was performed on rat brain sections essentially as previously described (*Manning et al., 2012*). All incubations and washes were at RT with slow agitation, unless stated otherwise. Briefly, free-floating sections were washed 3 × 5 min each in 0.1 M PB and 10 mM sodium azide. Sections were incubated in blocking buffer (10% goat serum in 0.1 M PB, 0.3% Triton X-100, and 10 mM sodium azide) for 1 hr. Immediately after blocking, sections were incubated with the primary antibody cocktail containing the candidate nAb BC supe and the mouse anti-Homer1 mouse IgG2b mAb L113/27 TC supe, both diluted 1:5 in blocking buffer, followed by overnight incubation at 4°C with slow agitation. Sections were then washed 3 × 10 min each in 0.1 M PB and incubated for 1 hr with a secondary antibody cocktail of anti-HA mouse IgG1 mAb 16B12 conjugated to Alexa Fluor 488 (ThermoFisher Cat# A-21287), to detect bound nAb, and goat anti-mouse IgG2b-subclass-specific Alexa Fluor 555-conjugated secondary antibody (ThermoFisher Cat# A21147) to detect the L113/27 mAb, both diluted 1:2000 in blocking buffer containing 500 ng/mL Hoechst 33258. After 3 × 10 min each washes in 0.1 M PB, sections were mounted and dried onto gelatin-coated slides, treated with 0.05% Sudan Black (EM Sciences Cat# 21610) in 70% ethanol for 2 min, extensively washed in water, and coverslipped using Prolong Gold (ThermoFisher Cat# P36930) mounting medium.

## Conventional diffraction-limited and super-resolution light microscopy

Images of fixed samples were acquired with an AxioCam MRm digital camera installed on a Zeiss AxioImager M2 microscope or with an AxioCam HRm digital camera installed on a Zeiss AxioObserver Z1 microscope with a 20X/0.8 NA plan-Apochromat objective or a 63X/1.40 NA plan-Apochromat oil immersion objective using an ApoTome structured illumination system for optical sectioning, with image acquisition controlled by Axiovision software (Zeiss, Oberkochen, Germany).

Images of the distribution of anti-Homer1 nAbs expressed in primary hippocampal cultures via lentivirus infection were acquired using a Nikon A1 Eclipse Ti confocal microscope with a 60x objective, operated by NIS Elements AR acquisition software (Nikon Instruments Inc, Melville, NY).

Live cell imaging of transfected hippocampal neurons was performed at the UC Davis MCB Imaging Facility using Total Internal Reflection Fluorescence (TIRF) microscopy as previously described (*Kirmiz et al., 2018a*). Live neurons cultured on glass bottom dishes were imaged in a physiological saline solution (4.7 mM KCl, 146 mM NaCl, 2.5 mM CaCl$_2$, 0.6 mM MgSO$_4$, 1.6 mM NaHCO$_3$. 0.15

mM NaH$_2$PO$_4$, 20 mM HEPES, pH 7.4) containing 8 mM glucose and 0.1 mM ascorbic acid. Cells were maintained at 37°C during the course of imaging with a heated stage and objective heater. Images were obtained with an Andor iXon EMCCD camera installed on a TIRF/widefield equipped Nikon Eclipse Ti microscope using a Nikon LUA4 laser launch with 405, 488, 561, and 647 nm lasers and a 100X PlanApo TIRF, 1.49 NA objective run with NIS Elements software (Nikon). Images were collected within NIS Elements as ND2 images. Time lapse movies, collected at 1 Hz, were collected for 1 min.

A super resolution ground-state depletion system (SR-GSD, Leica) based on stochastic single-molecule localization was used to generate super-resolution images. Images were obtained using a 160X HCX Plan-Apochromat (NA 1.43) oil-immersion lens and an EMCCD camera (iXon3 897; Andor Technology). For all experiments, the camera was running in frame-transfer mode at a frame rate of 100 Hz. A total of 35,000 images were used to construct the localization maps. Homer1 cluster sizes were determined using binary masks of the images in ImageJ software (NIH).

## Image analysis and statistics

All post-acquisition image analysis was performed using Fiji (*Schindelin et al., 2012*) except for analyses of images of anti-Homer1 nAb localization in primary hippocampal cultures via lentivirus, which was performed using Nikon Elements Analysis software. For line scan analyses of fluorescence intensity, the raw intensity values in the line scan analyses were collected within FIJI and normalized to the maximum value collected. The linear relationship between the fluorescence intensity of the signals was performed in Excel using the normalized values. Pearson's Correlation Coefficient (PCC) measurements were collected from ROIs manually drawn around dendrites (for nAbs targeting postsynaptic targets) or around the soma and proximal dendrites (for nAbs targeting AMIGO-1). Puncta size quantification was performed essentially as previous reported (*Bishop et al., 2018*; *Kirmiz et al., 2018a*; *Kirmiz et al., 2018b*). Briefly, images were identically background subtracted with a rolling ball radius of 10 pixels and converted into a binary image using automated local thresholding (*Bernsen, 1986*). Puncta sizes were then quantified using the 'analyze particles' function in Fiji. All data sets were imported into Prism Graphpad for statistical analysis and presentation. Statistical tests were performed as noted in each figure legend.

## Immunoblotting against brain samples

Immunoblots were performed on crude membrane fractions prepared from adult rat brain as previously described (*Trimmer, 1991*). Following determination of protein concentration by BCA assay (ThermoFisher Cat# 23227), 3 mg of RBM protein was loaded onto a single well that spanned the entire SDS-PAGE gel, electrophoresed to size-fractionate the proteins, and transferred onto a nitrocellulose membrane (BioRad Cat# 1620115). The nitrocellulose membrane was cut into 30 vertical strips so that each contained 100 μg of RBM protein. All remaining procedures were performed at RT. Strips were blocked for 1 hr in BLOTTO. Primary antibody incubation was performed using nAb BC supes diluted 1:2 in BLOTTO for 1 hr. Positive control antibodies were anti-Homer1 mouse IgG1 mAb L113/130 and anti-Gephyrin mouse IgG2a mAb L106/93, both used as TC supes diluted 1:2 in BLOTTO. Following primary antibody incubation, and 3 × 5 min each washes in BLOTTO, nAb strips were incubated for 1 hr in anti-HA rat IgG1 mAb 3F10 conjugated to HRP (Sigma-Aldrich Cat# 12013819001) diluted 1:2000 in BLOTTO, or HRP-conjugated goat anti-mouse IgG H+L (SeraCare Cat# 52200458) diluted 1:10,000 in BLOTTO. Following 3 × 5 min each washes in PBS, the chemiluminescent signal was generated by incubation in Western Lightning Plus ECL substrate (Fisher Scientific Cat# 509049325) and subsequently visualized on HyBlot CL film (Denville Scientific Cat# E3018).

## Acknowledgements

We thank Mikhail Melnik for expert assistance with the initial IHC analyses. We also thank Kimberley Nguyen and Grace Or Mizuno for help in preparation of cultured rat hippocampal neurons used in certain experiments. This work was supported by National Institutes of Health BRAIN Initiative Grants U01NS099714 and U24NS109113 to J S Trimmer.

## Additional information

### Funding

| Funder | Grant reference number | Author |
|---|---|---|
| National Institutes of Health | U01NS099714 | James S Trimmer |
| National Institutes of Health | U24NS109113 | James S Trimmer |

The funders had no role in study design, data collection and interpretation, or the decision to submit the work for publication.

### Author contributions

Jie-Xian Dong, Conceptualization, Resources, Formal analysis, Supervision, Validation, Investigation, Visualization, Methodology, Writing—original draft; Yongam Lee, Validation, Investigation, Methodology, Writing—review and editing; Michael Kirmiz, Data curation, Formal analysis, Validation, Investigation, Visualization, Methodology, Writing—review and editing; Stephanie Palacio, Camelia Dumitras, Validation, Investigation, Visualization, Methodology, Writing—review and editing; Claudia M Moreno, Conceptualization, Resources, Data curation, Formal analysis, Investigation, Visualization, Methodology, Writing—review and editing; Richard Sando, Conceptualization, Resources, Data curation, Formal analysis, Validation, Investigation, Visualization, Methodology, Writing—review and editing; L Fernando Santana, Resources, Data curation, Formal analysis, Supervision, Funding acquisition, Validation, Methodology, Writing—review and editing; Thomas C Südhof, Conceptualization, Resources, Data curation, Formal analysis, Supervision, Funding acquisition, Project administration, Writing—review and editing; Belvin Gong, Data curation, Formal analysis, Validation, Investigation, Writing—review and editing; Karl D Murray, Data curation, Supervision, Validation, Investigation, Methodology, Writing—review and editing; James S Trimmer, Conceptualization, Resources, Data curation, Formal analysis, Supervision, Funding acquisition, Validation, Investigation, Methodology, Project administration, Writing—review and editing

### Author ORCIDs

L Fernando Santana http://orcid.org/0000-0002-4297-8029
James S Trimmer https://orcid.org/0000-0002-6117-3912

### Ethics

Animal experimentation: All procedures involving llamas were performed at Triple J Farms of Kent Laboratories (Bellingham, WA) in strict accordance with the Guide for the Care and Use of Laboratory Animals of the NIH. All procedures involving rats were approved by the University of California, Davis, Institutional Animal Care and Use Committee (IACUC) under protocols 20485 and 21265 and were performed in strict accordance with the Guide for the Care and Use of Laboratory Animals of the NIH. All rats were maintained under standard light-dark cycles and allowed to feed and drink ad libitum. All procedures involving mice were approved by the Stanford University IACUC under protocol 18846 and were performed in strict accordance with the Guide for the Care and Use of Laboratory Animals of the NIH.

### Decision letter and Author response

Decision letter https://doi.org/10.7554/eLife.48750.021
Author response https://doi.org/10.7554/eLife.48750.022

## Additional files

### Supplementary files

• Supplementary file 1. Summary of corresponding mouse monoclonal antibody and nanobody immunolabeling results. Table lists the screening results from prior monoclonal antibody (mAb)

projects and the results from the corresponding nanobody screens. The numbers in parentheses in the final three columns of the nanobody rows represent the number of expected application-positive nanobodies obtained from the population of unique ELISA-positive nanobodies tested, based on percentages in mouse mAb projects.

DOI: https://doi.org/10.7554/eLife.48750.015

• Supplementary file 2. Summary of immunogens used for llama immunization. Table lists nanobody project target, fragment used for llama immunization and tag.

DOI: https://doi.org/10.7554/eLife.48750.016

• Supplementary file 3. Primers used for heavy chain repertoire cloning. Table lists primers used to amplify llama heavy chain variable regions.

DOI: https://doi.org/10.7554/eLife.48750.017

• Supplementary file 4. Non-nanobody antibodies used in this study. Table lists non-nanobody antibodies used throughout this study, the immunogen used in their development, the species and IgG subclass, the manufacturer and Antibody Registry/RRID information, and the figure in which each antibody was used.

DOI: https://doi.org/10.7554/eLife.48750.018

• Transparent reporting form

DOI: https://doi.org/10.7554/eLife.48750.019

## Data availability

All data generated or analyzed during this study are included in the manuscript and supporting files.

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
