## [Decision Letter]

[Editors’ note: this article was originally rejected after discussions between the reviewers, but the article was accepted after an appeal against the decision. The first decision letter following peer review is shown below]

Thank you for submitting your work entitled "A toolbox of nanobodies developed and validated for use as intrabodies and nanoscale immunolabels in brain neurons" for consideration by *eLife*. Your article has been reviewed by three peer reviewers, including Graeme W Davis as the Reviewing Editor and Reviewer #1, and the evaluation has been overseen by a Senior Editor.

The work was reviewed by three senior reviewers with experience directly relevant to the proposed resource. As you can appreciate from the reviews, there was general enthusiasm for the nature of the work, particularly the generation of reagents that could be used by the community. There were very few concerns, let alone suggestions regarding the generation of the tools or their demonstrated effectiveness and value. However, during the online discussion, a compelling argument was made that the actual resource that is being provided to the community is a relatively modest contribution and does not adhere to the *ELife* guidelines regarding a substantive contribution to the community. The decision hinged on the notion that the paper essentially contributed nanobodies to five antigens, which is not a large number and the nature of the pipeline lacks novelty in the current environment. Please see the comments of reviewer number 2. Further, in reference to the lack of novelty regarding the pipeline, it was noted that several papers have described similar pipelines. An older example is Fridy PC, Li Y, Keegan S, Thompson MK, Nudelman I, Scheid JF, Oeffinger M, Nussenzweig MC, Fenyö D, Chait BT, Rout MP. A robust pipeline for rapid production of versatile nanobody repertoires – published in Nat Methods. 2014 Dec;11(12):1253-60. There are others and a fair number of these are dutifully cited by the current authors. Ultimately, it is becoming standard to generate nanobodies. At least three companies offer the service. https://discover.abcore.com/nanobodies/. https://nano-tag.com/services. https://www.creative-biolabs.com. In conclusion, the reviewers are generally complimentary about the work and the presentation of the work. However, the novelty and impact of the resource was not deemed substantial enough to justify publication in *eLife*.

Reviewer #1:

The authors describe the generation and validation of nanobodies that can function as intrabodies. Several aspects are worth noting including the use of animal immunization rather than a synthetic approach. One of the intrabodies is validated for use in super-resolution microscopy. Ultimately, these will be valuable reagents for the community. There may be a general commitment toward further reagent generation in the future. In general, as a community resource, it seems to me that the paper succeeds.

Reviewer #2:

In this Tools and Resources paper, Trimmer and colleagues describe generation of nanobodies to 5 neuronal proteins, and their use for immunolabeling neurons. In brief, they generated and pooled recombinant fragments of the 5 proteins, immunized a llama with the pool, generated a library from lymphocyte RNA, isolated antibodies by phage display, transferred the cDNAs to mammalian expression vectors containing a GFP tag, and screened them by expression in heterologous cells. They then introduced a subset of them into neurons by electroporation or lentiviral infection, and used them as "intrabodies" to localize endogenous proteins at short and long intervals. Finally, they showed that a subset of the nanobodies could be used to label antigen in brain sections and on immunoblots.

The manuscript is clear, complete and very-well illustrated. On the other hand, I do not think it reports an advance of sufficient novelty or utility to merit publication in *eLife*. Editorial guidance provided to the reviewers states that Tools and Resources papers should report "significant technological or methodological advances, genomic or other datasets, collections of biological resources, software tools, and so on." In this case, all of the methods for generating and using nanobodies and intrabodies have been described previously by multiple groups; many of the relevant papers are cited, so I won't list them here. In this respect, it differs from this group's recent paper on generating recombinant monoclonal antibodies (Andrews et al., *eLife*, 2019), which introduces a suite of methods that had been little used and are likely to be broadly useful. The "biological resource" is the set of new nanobodies, but with only 5 targets, it is quite small in comparison to previous datasets from this and other groups.

Reviewer #3:

Dong and colleagues have developed a toolbox of nanobodies that can be used to recognize a collection proteins expressed at excitatory and inhibitory synapses as well as at the ER-PM interface. They perform a series of experiments testing/validating the utility of these nAbs by expressing them in COS cells and neurons as well as purifying them for use in staining in fixed tissue. This tool kit will be generally useful and provides a platform for future derivative uses.

I recommend publication, pending addressing a few concerns.

1) The authors show in Figure 2 that with the "long term" expression of homer nAb, puncta can still be resolved. Is this the case for the other nAbs? This is really the crux of the utility of the tools and should be shown.

2) Data illustrated in Figures 3 and 4 need quantification or some sort.

3) A table with rank ordering or quality assessment of the efficacy of all nAbs would be useful. It is transparent, but not necessarily user friendly, to provide a collection of nAbs without some ranking of which one(s) people should try first.

4) The Arnold lab has made intrabodies recognizing PSD-95 and gephryin. This should be referenced and discussed. How is this strategy better/different?

---

## [Author Response]

[Editors’ note: the author’s appeal following peer review and the first decision letter follows.]

The work was reviewed by three senior reviewers with experience directly relevant to the proposed resource. As you can appreciate from the reviews, there was general enthusiasm for the nature of the work, particularly the generation of reagents that could be used by the community. There were very few concerns, let alone suggestions regarding the generation of the tools or their demonstrated effectiveness and value. However, during the online discussion, a compelling argument was made that the actual resource that is being provided to the community is a relatively modest contribution and does not adhere to the eLife guidelines regarding a substantive contribution to the community. The decision hinged on the notion that the paper essentially contributed nanobodies to five antigens, which is not a large number and the nature of the pipeline lacks novelty in the current environment. Please see the comments of reviewer number 2. Further, in reference to the lack of novelty regarding the pipeline, it was noted that several papers have described similar pipelines. An older example is Fridy PC, Li Y, Keegan S, Thompson MK, Nudelman I, Scheid JF, Oeffinger M, Nussenzweig MC, Fenyö D, Chait BT, Rout MP. A robust pipeline for rapid production of versatile nanobody repertoires – published in Nat Methods. 2014 Dec;11(12):1253-60. There are others and a fair number of these are dutifully cited by the current authors. Ultimately, it is becoming standard to generate nanobodies. At least three companies offer the service. https://discover.abcore.com/nanobodies/. https://nano-tag.com/services. https://www.creative-biolabs.com. In conclusion, the reviewers are generally complimentary about the work and the presentation of the work. However, the novelty and impact of the resource was not deemed substantial enough to justify publication in eLife.Reviewer #2:[…] The manuscript is clear, complete and very-well illustrated. On the other hand, I do not think it reports an advance of sufficient novelty or utility to merit publication in eLife. Editorial guidance provided to the reviewers states that Tools and Resources papers should report "significant technological or methodological advances, genomic or other datasets, collections of biological resources, software tools, and so on." In this case, all of the methods for generating and using nanobodies and intrabodies have been described previously by multiple groups; many of the relevant papers are cited, so I won't list them here. In this respect, it differs from this group's recent paper on generating recombinant monoclonal antibodies (Andrews et al., eLife, 2019), which introduces a suite of methods that had been little used and are likely to be broadly useful. The "biological resource" is the set of new nanobodies, but with only 5 targets, it is quite small in comparison to previous datasets from this and other groups.Reviewer #3:[…] I recommend publication, pending addressing a few concerns.1) The authors show in Figure 2 that with the "long term" expression of homer nAb, puncta can still be resolved. Is this the case for the other nAbs? This is really the crux of the utility of the tools and should be shown.2) Data illustrated in Figures 3 and 4 need quantification or some sort.3) A table with rank ordering or quality assessment of the efficacy of all nAbs would be useful. It is transparent, but not necessarily user friendly, to provide a collection of nAbs without some ranking of which one(s) people should try first.4) The Arnold lab has made intrabodies recognizing PSD-95 and gephryin. This should be referenced and discussed. How is this strategy better/different?

I am writing to appeal the editorial decision on our manuscript on the basis that:

1) To date, only one nanobody has been generated against a neuronal target (synapsin), and in this effort nanobodies were only evaluated for use as immunolabels and not for use as intrabodies. Here we developed a large collection of nanobodies against five targets and evaluated them for both uses.

2) To date, a very small number of intrabodies of any kind have been developed against neuronal targets for neuroscience research, most prominently the two monobodies/FingRs (against PSD-95 and Gephyrin) developed in Don Arnold’s lab and published in Neuron, and that have gained widespread use, showing the overall need for such reagents.

3) While the development and screening pipeline we used, like all pipelines, employed individual components similar to those used in published work, this is the first time anyone has used them in this novel combination to screen a large set of nanobodies or other small format single chain binders against endogenous targets for those that work as both intrabodies and immunolabels, and that can be used in brain samples.

4) The value of these nanobodies is evident by the number of requests we have received for these reagents due to the bioRXIV posting that was suggested by *eLife*.

5) The value of this resource is further underscored in that our work was selected by Walter Koroshetz, Director of NINDS, to be presented in a mini-symposium at the 2019 SFN meeting whose focus is to highlight important research resources generated under the BRAIN Initiative. It will also be highlighted in a brief minisymposium article in the Journal of Neuroscience.